# Analysis of PARP inhibitor toxicity by multidimensional fluorescence microscopy reveals mechanisms of sensitivity and resistance

Jone Michelena[1], Aleksandra Lezaja[1], Federico Teloni[1], Thomas Schmid[1], Ralph Imhof[1] & Matthias Altmeyer [1]

Exploiting the full potential of anti-cancer drugs necessitates a detailed understanding of their cytotoxic effects. While standard omics approaches are limited to cell population averages, emerging single cell techniques currently lack throughput and are not applicable for compound screens. Here, we employed a versatile and sensitive high-content microscopy-based approach to overcome these limitations and quantify multiple parameters of cytotoxicity at the single cell level and in a cell cycle resolved manner. Applied to PARP inhibitors (PARPi) this approach revealed an S-phase-specific DNA damage response after only 15 min, quantitatively differentiated responses to several clinically important PARPi, allowed for cell cycle resolved analyses of PARP trapping, and predicted conditions of PARPi hypersensitivity and resistance. The approach illuminates cellular mechanisms of drug synergism and, through a targeted multivariate screen, could identify a functional interaction between PARPi olaparib and NEDD8/SCF inhibition, which we show is dependent on PARP1 and linked to PARP1 trapping.

[1] Department of Molecular Mechanisms of Disease, University of Zurich, CH-8057 Zurich, Switzerland. These authors contributed equally: Aleksandra Lezaja, Federico Teloni. Correspondence and requests for materials should be addressed to M.A. (email: matthias.altmeyer@uzh.ch)

Following two seminal publications in 2005 demonstrating greatly increased sensitivity of *BRCA1/2* mutant cancer cells to poly(ADP-ribose) polymerase (PARP) inhibition[1,2], PARP inhibitors (PARPi) have been extensively tested for their potential as single therapeutic agents based on the concept of tumor-specific synthetic lethality[3–5]. In 2014, olaparib (Lynparza, AstraZeneca) was approved by the European Medicines Agency (EMA) and the US Food and Drug Administration (FDA) for the treatment of *BRCA1/2* mutant ovarian cancers[6]. Several additional PARPi, including talazoparib, niraparib, rucaparib and veliparib, are currently in late phase clinical trial development or have recently been approved[7,8]. PARPi target PARP enzymes (mainly PARP1 and PARP2), which are DNA damage sensors that catalyze the formation of negatively charged poly(ADP-ribose) (PAR) chains to regulate protein assemblies and tune chromatin dynamics in response to genotoxic stress[9–13]. Notably, PARPs are not only implicated in maintaining genome stability, but also have functions in various other cellular contexts, including chromatin remodeling, transcription, and mRNA processing, and they play important roles in cellular differentiation, embryonic development, inflammation, metabolism, cancer development, and aging[14–17]. While the mechanisms of action of PARPi are incompletely understood and likely involve multiple molecular events, including impaired recruitment of repair proteins to sites of DNA damage, deregulated replication fork reversal and reduced fork stability, as well as PARP trapping and the formation of toxic PARP-DNA complexes that may give rise to replication-associated DNA damage[18–25], it has become clear that an exquisite vulnerability to PARPi exists in cells with compromised homologous recombination (HR) capacity[26]. This synthetic lethal relationship between PARPi and compromised HR function can explain the sensitivity of *BRCA1/2* mutant cells to PARPi, and strategies are currently being explored to identify predictive biomarkers for PARPi sensitivity[26].

Besides the current lack of strong predictive biomarkers for PARPi responses, recently emerging mechanisms of PARPi resistance in advanced disease complicate their clinical use. These include regained HR capacity through restoration of BRCA1/2 function or through compensatory loss of functional antagonists, reduced drug uptake through up-regulation of the P-glycoprotein drug efflux transporter, and loss of PARP1 expression[27,28].

Despite the broad interest in PARPi and their clinical potential, how inhibition of PARP enzymes translates into cell death and how cells can overcome PARPi sensitivity is currently not well understood. In light of the clinical and pre-clinical challenges to understand PARPi functions and evaluate their cellular effects, experimental systems to assess PARPi toxicity at multiple levels in a sensitive and quantitative manner are needed. Such systems would enable the assessment of cellular mechanisms of PARPi sensitivity and resistance and further reveal how PARPi resistance might be overcome, e.g., through combined drug treatments. Current methods employed to evaluate PARPi toxicity mostly rely on long-term cell proliferation and clonogenic survival assays, manual assessment of PARPi-induced DNA damage markers such as γH2AX or RAD51 in relatively small cohorts of cells, or biochemical cell fractionation for the detection of chromatin-bound PARP1[29–34]. Despite all benefits, these approaches are typically either time consuming, have limited sensitivity, are not well suited for screening purposes, or focus on single parameters of the cellular response to PARPi. Moreover, cell-to-cell variation in PARPi responses is often not accounted for and cannot be assessed in measurements of cell population averages. This extends to cell cycle phase-specific responses, which are common to many cytotoxic agents, and which are easily lost in cell population averages of asynchronously growing cells. High-throughput single-cell assays can discern sub-population-specific responses and thereby reveal the dynamics of cellular responses to drug perturbations[35–38]. More specifically, high-content microscopy can be used to stage cells according to their position in the cell cycle and to correlate cell cycle dynamics with cellular stress responses[39–46]. In light of the limitations associated with current tools used to dissect PARPi responses, we aimed to exploit the power of single-cell analyses for detailed multidimensional characterization of the cell biology underlying PARPi toxicity. Building on the high resolution and throughput of current fluorescence microscopes, we used a flexible experimental workflow that discerns cell cycle phase specific responses to PARPi and quantifies relevant parameters of PARPi cytotoxicity in a sensitive and reliable manner. It faithfully detects differences in PARP1 trapping potential and toxicity of a panel of clinically relevant PARPi, and reveals mechanisms of PARPi sensitivity and resistance in BRCA1/2- and FANCD2-deficient cells and upon down-regulation of PARP enzymes, respectively. Moreover, it is highly versatile and can be used to discern cellular responses to a broad range of chemotherapeutic drugs and to illuminate synergistic effects between PARPi and other genotoxins, even upon short-term treatments. Based on these analyses, we reveal how PARPi and inhibition of the DNA damage response kinase ATR synergize by promoting replication-born lesions to be transmitted to mitosis where they give rise to catastrophic chromosomal damage. We also reveal an unexpected role of the PARP antagonist PARG in mediating PARPi-toxicity. Finally, a targeted high-content imaging screen led to the identification of an interaction between PARPi and inhibition of the NEDD8/SCF machinery, which we show is mechanistically linked to PARP1 trapping and can be rescued by PARP1 down-regulation. Assessing cellular responses to PARPi by multidimensional fluorescence microscopy thus provides a sensitive, reliable, easy-to-use and cost-efficient means to assess PARPi toxicity in low and high throughput and to reveal mechanisms of PARPi sensitivity and resistance.

## Results

**Quantitative image-based cytometry to measure PARPi toxicity.** To evaluate PARPi effects on proliferating cancer cell populations at the single-cell level, we employed automated fluorescence microscopy in combination with software-assisted image analysis. We treated asynchronously growing U-2 OS cells with the PARPi olaparib (Lynparza) and, using optimized immunofluorescence staining conditions for quantitative image-based cytometry (QIBC)[43–46], we stained for γH2AX (Serine139-phosphorylated histone H2AX) and DNA content as a cell cycle marker. We then used automated multi-position microscopy to acquire image information of several thousand cells per condition. This enabled us to monitor effects of PARPi at the single cell level and in a cell cycle resolved manner, thus illuminating the dynamics and the cell cycle phase specificity of the response to PARPi. Image-based cell cycle staging revealed a PARPi-induced increase in γH2AX specifically in a sub-population of cells with intermediate DNA content, while cells with low (2C) and high (4C) DNA content did not show any measurable increase in PARPi-evoked DNA damage response signaling (PARPi-DDR) (Fig. 1a). Combined PARPi-DDR measurements with 5-ethynyl-2'-deoxyuridine (EdU) labeling of DNA replication confirmed that the PARPi-induced γH2AX formation was indeed confined to the S-phase population (Supplementary Figure 1a, b). Thus, PARPi leads to S-phase-specific DNA damage signaling, which can be readily detected by QIBC independent of cell synchronization or other chemical or physical perturbations. This result was confirmed in the non-cancerous epithelial cell line RPE1 (Supplementary Figure 1c). To compare the sensitivity of our

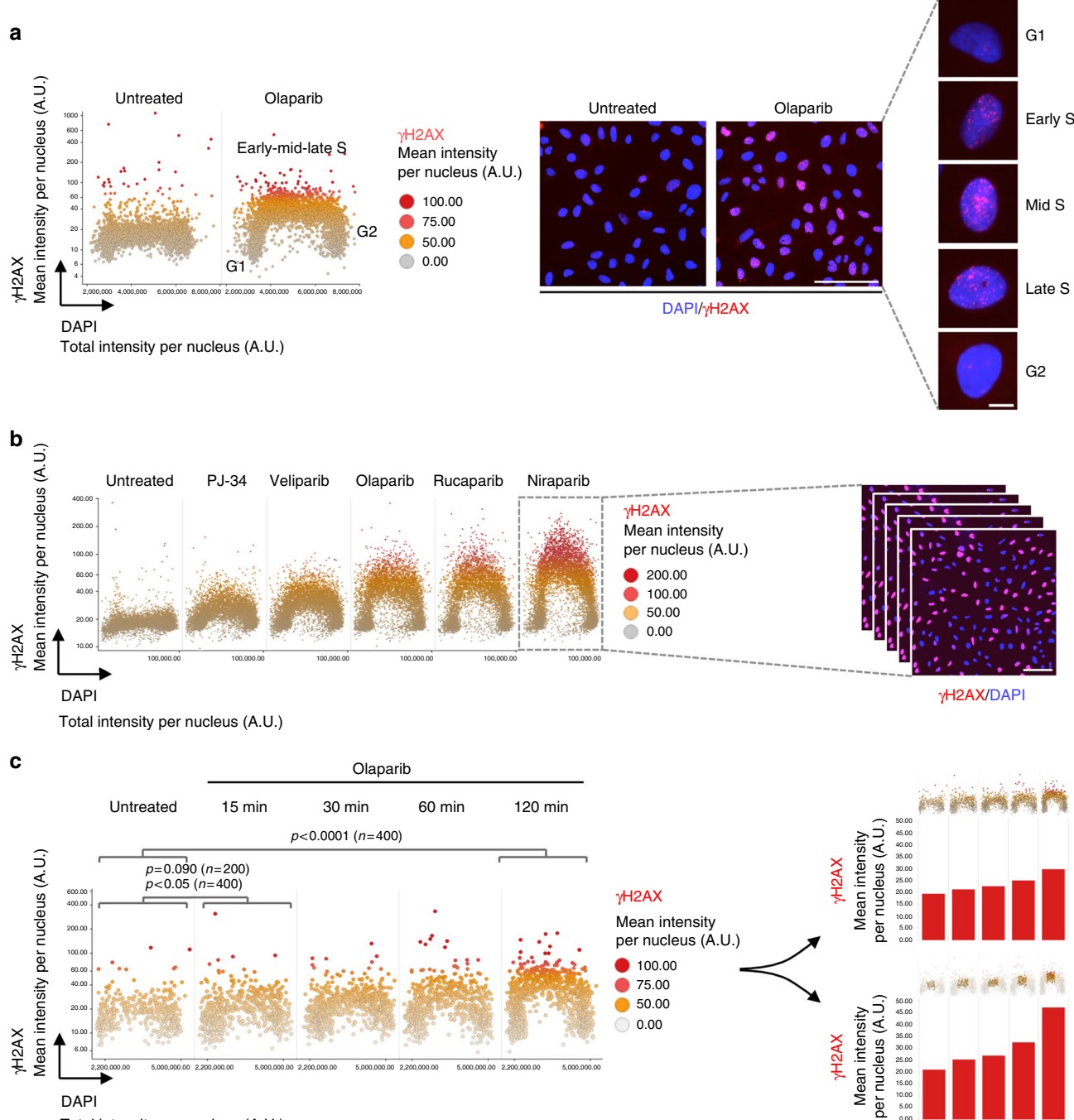

**Fig. 1** Quantitative analysis of PARP inhibitor toxicity by multidimensional fluorescence microscopy. **a** Asynchronously growing populations of adherent U-2 OS cells were treated with 10 μM olaparib for 4 h, fixed and stained for DNA content (DAPI) and the genotoxic stress marker γH2AX. Scatter plots depict mean γH2AX and total DAPI intensities per nucleus. Each dot represents a single cell. Representative images of the untreated and olaparib-treated cell populations and individual cells are shown on the right. **b** U-2 OS cells were treated with the indicated PARPi (all 10 μM) for 4 h and stained for γH2AX and DNA content as before. γH2AX levels are shown as a function of cell cycle progression. A representative picture of niraparib-treated cells is shown on the right. **c** Cells were treated as indicated, pre-extracted on ice in 0.2% Triton X-100 for 2 min prior to fixation, and stained for γH2AX and DNA content. γH2AX was quantified as a function of cell cycle progression. On the right, bar charts depict γH2AX levels as cell population averages from the same time-course either focused on the whole cell population (top right) or on cells in mid S-phase (bottom right). Note that focusing the analysis on specific sub-populations only (e.g., excluding cells in G1 and G2) greatly enhances the dynamic range of the analysis. p-values were obtained by Mann–Whitney test. Color codes as defined in the figure panels. Scale bars, 100 μm (large image fields) and 10 μm (single cell images)

experimental system to alternative approaches, we exposed cells to olaparib or treated them with ionizing radiation (IR) and extracted histones for analysis by western blot. While IR-induced phosphorylation of H2AX could be clearly detected, this was not the case for the PARPi-treated samples (Supplementary

Figure 1d). Thus, QIBC is superior in sensitivity, presumably in part due to its ability to demerge sub-population specific responses. Indeed, when we synchronized cells in G1 and released them into S-phase prior to histone extraction we were able to detect a moderate increase in γH2AX upon PARPi

(Supplementary Figure 1e, f). Also flow cytometry revealed olaparib-induced γH2AX signaling in S-phase cells (Supplementary Figure 1g). However, compared to microscopy-based measurements the sensitivity of both western blot and flow cytometry was extenuated and the lack of high-resolution cell images represents an additional limitation that precludes analyses of PARPi-induced changes at the subcellular level (see below).

Having validated the approach, we assessed a larger panel of clinically relevant PARPi. Besides olaparib, several potent PARPi are currently being tested in the clinics, including the third generation PARPi veliparib (ABT-888, Abbvie), rucaparib (AGO14699, Clovis) and niraparib (MK4827, Tesaro)[7,8]. We also included PJ-34, a second generation PARPi that has been used for blocking PARP catalytic activity for more than 15 years[47]. All compounds resulted in clearly detectable S-phase specific γH2AX signaling after just 4 h of exposure, with niraparib having the strongest and PJ-34 the weakest effect (Fig. 1b). While single immunofluorescence images reveal the vast cellular heterogeneity in the response to PARPi, the high-content imaging-based cell cycle staging precisely attributes the differential response to specific cell cycle phases (Fig. 1b, exemplified for niraparib). Remarkably, the observed differences in DNA damage signaling (niraparib >> rucaparib >> olaparib >> veliparib >> PJ-34) closely matched previously reported $IC_{90}$ values for these compounds[7,22,30] and they were well aligned with cell proliferation measurements performed by high-content time-lapse imaging over 72 h in absence or presence of PARPi (Supplementary Figure 1h). Thus, QIBC-based short-term measurements of PARPi-DDR in S-phase cells may be used as a rapid readout for sensitivity to different PARPi. To fully exploit the predictive capability of the system we aimed to further enhance its sensitivity. Detergent-based pre-extraction of cells prior to fixation indeed enhanced the PARPi-induced γH2AX signal (Supplementary Figure 1i) and allowed us to detect S-phase DNA damage signaling as early as 15 min after olaparib addition (Fig. 1c). Significance testing of these data suggests that a comparison of 400 cells per condition was enough to discriminate PARPi-treated from untreated cells with reasonable ($p < 0.05$ at 15 min, Mann–Whitney test) to high confidence ($p < 0.0001$ at 2 h). Importantly, focusing the analysis on sub-populations, e.g. on S-phase cells only, greatly enhanced the dynamic range of the quantitation (Fig. 1c, compare upper right panel for whole cell populations to lower right panel focused on mid S-phase cells). Taken together, these results demonstrate that quantitative analysis of PARPi-DDR by fluorescence microscopy is exquisitely sensitive and can predict varying degrees of toxicity of clinically relevant PARPi in short-term experiments.

Given that cancer cells with defects in the homologous recombination (HR) pathway are hypersensitive to PARPi[5], we next tested whether we could detect HR intermediates in PARPi-treated cells. The RAD51 recombinase forms filaments on damaged DNA to pair with homologous DNA sequences and initiate HR repair. These HR intermediates can be detected microscopically as nuclear RAD51 foci upon DNA breakage or PARPi exposure[48,49]. We observed a time-dependent increase in RAD51 foci as early as 1 h after PARPi treatment, which was most prominent in mid S-phase cells (Supplementary Figure 2a). RAD51 foci were found in cells containing the highest amount of γH2AX (Supplementary Figure 2b), and in these cells we detected a decrease in replication speed, which was associated with a delay in S-phase progression (Supplementary Figure 2c). As expected, RAD51 foci formation was completely dependent on the HR factors BRCA1 and BRCA2 (Supplementary Figure 2d). We next tested whether the QIBC-based PARPi-DDR approach could faithfully recapitulate known clinically relevant mechanisms of PARPi sensitivity, specifically loss of BRCA1/2[1,2] and Fanconi anemia (FA) genes[50,51]. Although the

initial γH2AX response to PARPi was comparable between BRCA1/2-depleted and control cells, knockdown of BRCA1 or BRCA2 markedly increased γH2AX in S/G2 after 16 and 48 h of PARPi treatment (Fig. 2a, b). Image-based cell cycle staging of asynchronous cells further revealed that loss of BRCA1/2 led to higher amounts of mitotic DNA damage as demonstrated by γH2AX-positive condensed mitotic chromosomes (Supplementary Figure 3a), in line with recent work[52]. Similar to BRCA1/2 deficiency, loss of FANCD2 resulted in higher amounts of DNA damage (Fig. 2c and Supplementary Figure 2e for knockdown controls for BRCA1, BRCA2 and FANCD2). We also compared a BRCA1-mutated, PARPi-sensitive triple negative breast cancer (TNBC) cell line, MDA-MB436, to a BRCA-proficient TNBC cell line, HCC1143[53], and observed PARPi-induced γH2AX in MDA-MB436 but not in HCC1143 cells, consistent with their PARPi-sensitivity status (Fig. 2d).

Interestingly, recent work established that the cellular metabolite and ubiquitous environmental toxin formaldehyde induces a BRCA haploinsufficiency by accelerating degradation of BRCA2 and, to a lesser extent, BRCA1 and RAD51, which in turn promotes genome instability[54]. Based on these findings, we tested whether formaldehyde exposure would also cause PARPi hypersensitivity in otherwise BRCA1/2 wild-type cells. Indeed, formaldehyde treatment resulted in a marked increase in PARPi-induced S-phase-specific DNA damage signaling, consistent with compromised BRCA function and suggesting that PARPi and formaldehyde functionally synergize (Supplementary Figure 3b).

We next tested whether we could monitor cellular responses to a wider variety of genotoxic agents. First, we followed the induction of DNA damage signaling upon ionizing radiation (IR) and its gradual decay over time, unveiling a cell cycle independent induction of γH2AX (Supplementary Figure 3c). We then monitored single and combined treatments of chemotherapeutically relevant genotoxic agents such as IR, the topoisomerase I poison camptothecin (CPT), the methylating agent temozolomide (TMZ), and the PARPi olaparib, to assess the potential therapeutic value of different combination therapies. These experiments revealed treatment-specific γH2AX patterns, which demonstrates their differential effect on inducing the DDR, and provided a detailed picture of how IR, CPT and TMZ synergize with PARPi, resulting in a PARPi-induced S-phase specific increase of DNA damage signaling (Supplementary Figure 3d). As before, focusing the analysis only on S-phase cells increased the sensitivity of the readout (Supplementary Figure 3d, lower panels). Thus, QIBC-based PARPi-DDR measurements can reveal cytotoxic interactions with a variety of chemotherapeutic agents and discern their relative strengths.

**Cell cycle resolved quantification of PARP trapping.** PARPi-induced cytotoxicity has been linked to an additional feature of PARPi, namely their potential to lock PARP enzymes in an inactive state and trap PARP enzymes and associated proteins on chromatin, generating cytotoxic protein-DNA complexes[25]. Consistent with this model, PARPi toxicity is relieved in the absence of PARP enzymes[22,23]. PARP trapping has been detected biochemically by chromatin fractionation experiments from bulk cell populations exposed to the alkylating agent methyl methanesulfonate (MMS)[22,23]. We aimed to test whether high-content single cell imaging could provide an alternative means to evaluate and quantify PARP trapping in a manner that would allow us to directly correlate PARP trapping and PARPi-induced DNA damage signaling in the same cells. By including a pre-extraction step prior to fixation to specifically detect chromatin-bound proteins[46] we indeed observed a marked increase in the chromatin retention of PARP1 when cells were exposed to MMS and the PARPi olaparib for 4 h (Fig. 3a). In the same cells, we measured γH2AX signaling (Fig. 3b) and EdU

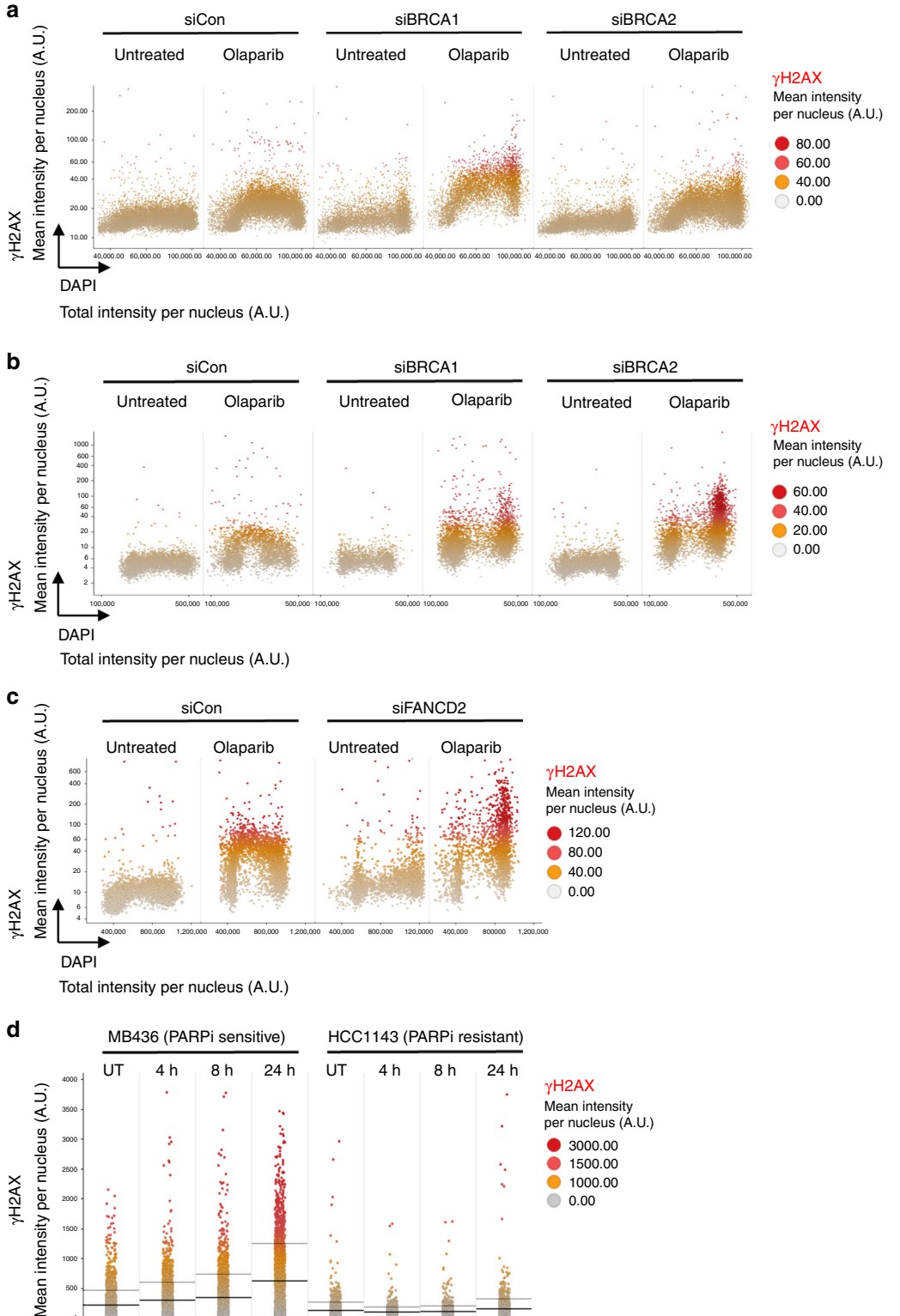

**Fig. 2** Defects in HR factors confer hypersensitivity to PARPi. **a** U-2 OS cells were transfected with negative control siRNA or siRNAs against BRCA1 or BRCA2, treated with 10 μM olaparib for 16 h, and stained for γH2AX and DNA content. **b** Cells were transfected as in **a** and treated as indicated with 10 μM olaparib for 48 h. Cell cycle resolved γH2AX profiles are shown. **c** Cells were transfected with negative control siRNA or siRNA against FANCD2, treated as indicated with 10 μM olaparib for 48 h, and cell cycle resolved γH2AX profiles are shown. **d** MDA-MB436 and HCC1143 cells were treated with 10 μM olaparib as indicated and stained for γH2AX. Horizontal lines represent population averages ± s.d. Color codes as defined in the figure panels

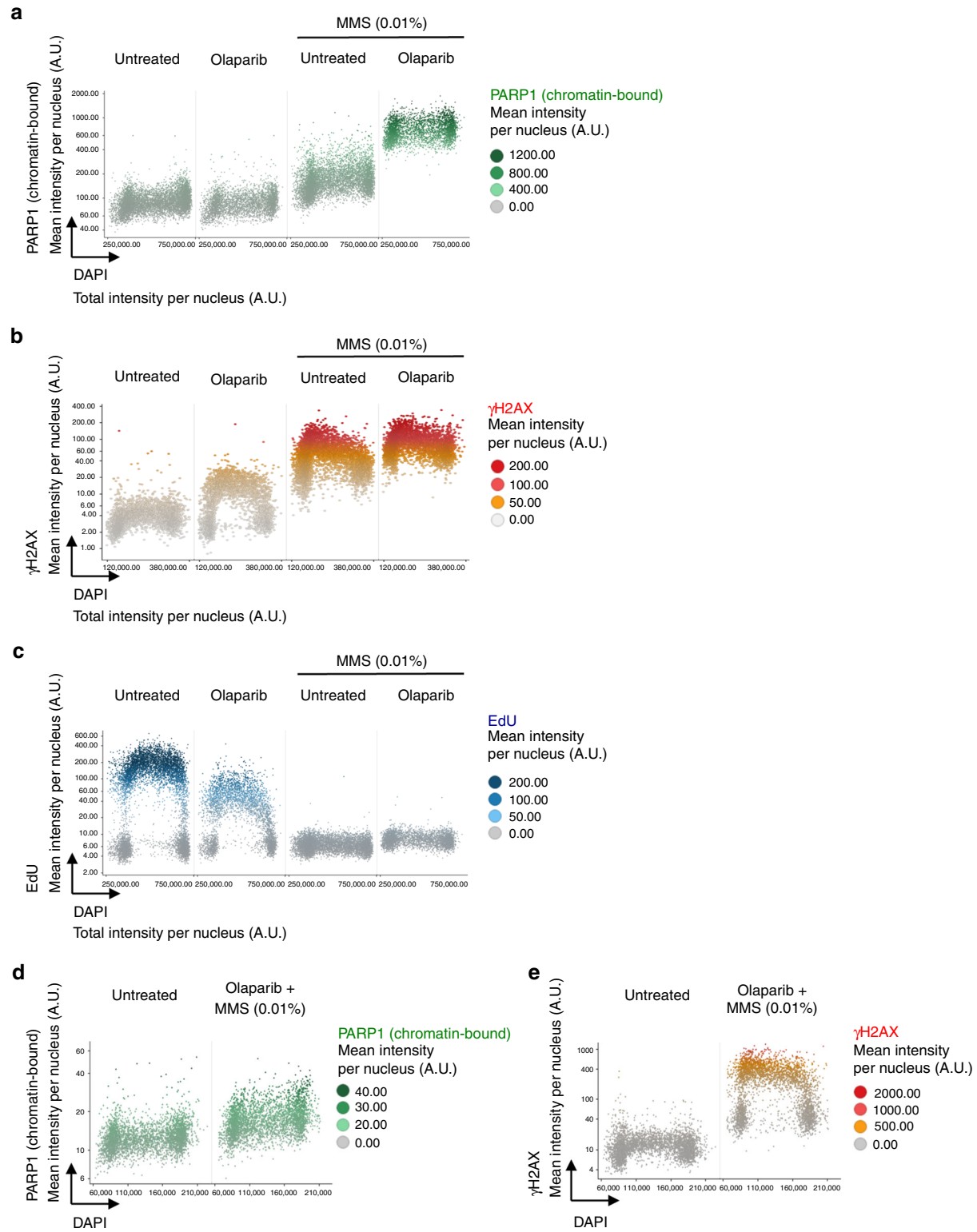

**Fig. 3** Image-based quantification of PARPi-induced PARP1 trapping. **a** U-2 OS cells were treated with 10 μM olaparib and 0.01% MMS for 4 h as indicated, pre-extracted on ice in 0.2% Triton X-100 for 2 min to remove soluble, non-chromatin-bound proteins, and stained for PARP1, γH2AX and DNA content. Chromatin-bound PARP1 levels were quantified and are depicted as a function of cell cycle progression. **b** For the same cell populations, γH2AX levels were quantified and are depicted as a function of cell cycle progression. **c** For the same cell populations analyzed in **a** and **b** two-dimensional cell cycle profiles based on DAPI/EdU are shown. **d** Cells were treated with 10 μM olaparib and 0.01% MMS for 1 h, pre-extracted on ice in 0.2% Triton X-100 for 2 min to remove soluble, non-chromatin-bound proteins, and stained for PARP1, γH2AX and DNA content. Chromatin-bound PARP1 levels were quantified and are depicted as a function of cell cycle progression. **e** For the same cell populations, γH2AX levels were quantified and are depicted as a function of cell cycle progression. Color codes as defined in the figure panels

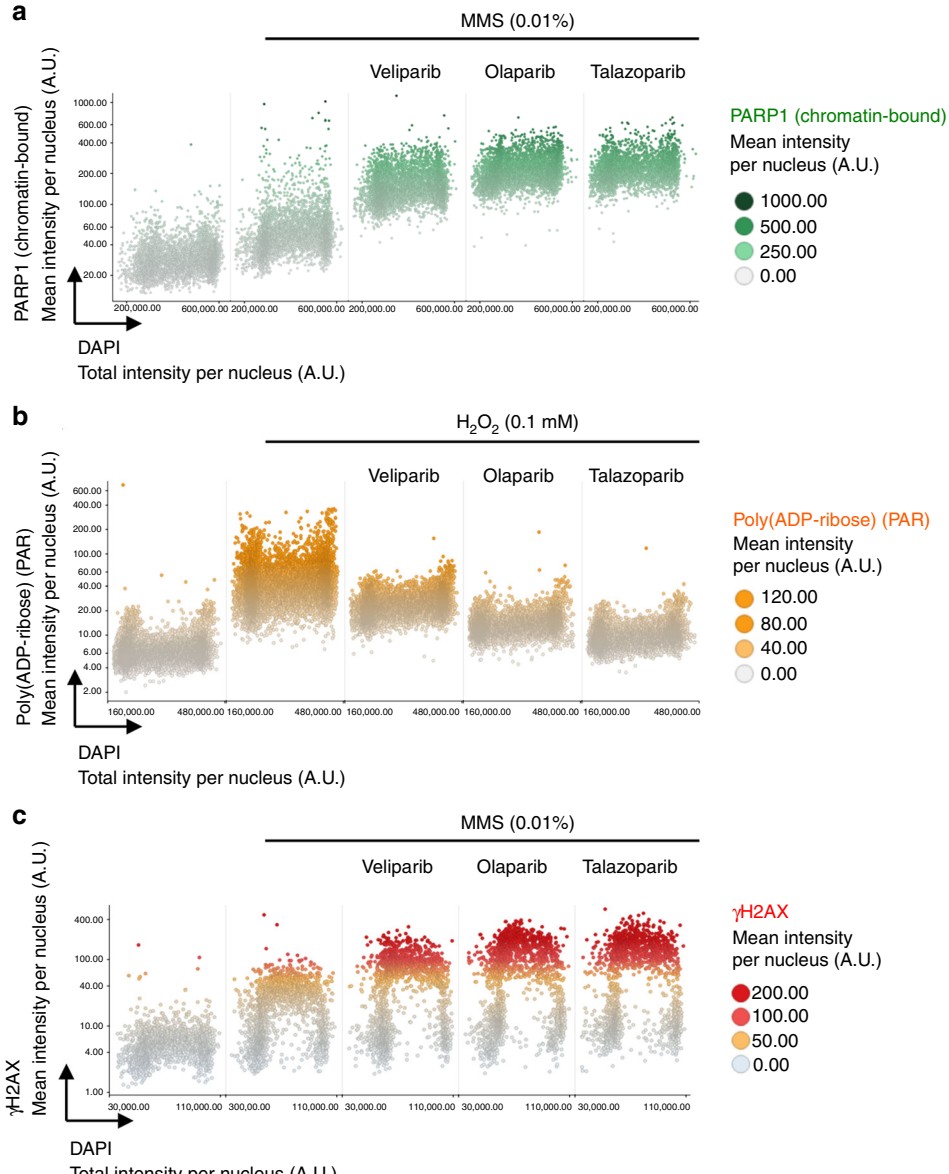

**Fig. 4** Differential PARP trapping and PARP inhibition by a panel of PARPi. **a** U-2 OS cells were treated with different PARPi (10 μM) for 4 h as indicated, pre-extracted on ice with 0.2% Triton X-100 for 2 min to remove soluble, non-chromatin-bound proteins, and stained for PARP1 and DNA content. Chromatin-bound PARP1 levels were quantified and are depicted as a function of cell cycle progression. **b** Cells were pre-incubated with different PARPi for 30 min as indicated, treated with H₂O₂ for 10 min to induce PAR formation, and stained for PAR and DNA content. Cellular PAR levels were quantified and are depicted as a function of cell cycle progression. **c** Cells were treated with different PARPi (10 μM) for 1 h as indicated, pre-extracted on ice in 0.2% Triton X-100 for 2 min and stained for γH2AX and DNA content. γH2AX levels were quantified and are depicted as a function of cell cycle progression. Color codes as defined in the figure panels

incorporation (Fig. 3c). Interestingly, while PARP1 trapping was independent of the cell cycle, the initial DNA damage signaling after 1 h treatment occurred primarily in S-phase cells, supporting the notion that PARP1 trapping occurs in all cell cycle phases but causes most problems during DNA replication (Fig. 3d, e). The detected signals for chromatin-bound PARP1 were specific (Supplementary Figure 4a), and biochemical fractionation confirmed the QIBC-based PARP1 trapping results, although with lower sensitivity and quantitative power, and without the possibility to directly correlate PARP1 trapping to γH2AX formation at the single cell level (Supplementary Figure 4b). We next tested three different PARPi (veliparib, olaparib, and talazoparib), which vary in their cytotoxicity and PARP trapping potential[29]. While all three inhibitors resulted in greatly enhanced chromatin retention of PARP1 in the presence of

MMS, veliparib had the weakest and talazoparib the strongest effect (Fig. 4a). The degree of PARP1 trapping (talazoparib >> olaparib >> veliparib) correlated with the degree of PARP inhibition as revealed by measurements of hydrogen peroxide (H₂O₂) induced poly(ADP-ribose) (PAR) formation (Fig. 4b) and with the amount of S-phase-specific DNA damage signaling (Fig. 4c).

It was previously suggested that synergy between TMZ and PARPi is based on PARP trapping in addition to PARP catalytic inhibition, while synergy between CPT and PARPi is primarily due to PARP catalytic inhibition and does not require PARP trapping[23,25]. In line, we indeed observed elevated PARP1 chromatin retention upon PARPi/TMZ but not PARPi/CPT (Supplementary Figure 4c, d), consistent with the differential strength of synergism by these drug combinations (Supplementary Figure 3d).

Exposure to either MMS or the known PARP activator $H_2O_2$ led to elevated levels of detergent-resistant, chromatin-bound PARP1 upon PARPi (Supplementary Figure 5a, b). However, we were unable to detect significant increases in PARP trapping upon PARPi alone. While chromatin trapping of PARP1 and associated proteins may occur in the presence of PARPi alone below our detection limit, it is also possible that PARP trapping occurs primarily under conditions when PARP enzymes are highly activated and at the same time catalytically blocked. Accordingly, genotoxins forming PARP-activating lesions might generally work more via PARP trapping in conjunction with PARPi as compared to drugs that do not result in high levels of PARP activation. To consolidate this point and to test whether the high-content imaging-based PARP1 trapping assay was compatible with high-throughput screening, we performed a proof-of-concept combinatorial drug screen in technical quadruplicates at two different time-points in multiwell format and simultaneously assessed chromatin-bound PARP1, γH2AX, and EdU. This revealed both similarities and differences between the different drug combinations at the level of PARP1 trapping, DNA damage signaling and DNA replication (Supplementary Figure 5c). The technical replicates showed overall consistent results (Supplementary Figure 5d, left panels), and the screen revealed unexpected differences in the kinetics of PARP1 trapping and γH2AX induction when comparing MMS and $H_2O_2$, although both affected EdU incorporation in a very similar manner (Supplementary Figure 5d, middle panels). It also provided further insights into the synergy between TMZ and olaparib, which we found to functionally interact at all measured parameters (Supplementary Figure 5d, right panels). In all tested conditions, PARP1 trapping occurred in a cell cycle-independent manner. PARP trapping measurements by QIBC, combined with additional markers of drug cytotoxicity, thus represent a screening-compatible experimental pipeline to interrogate cellular responses to single drugs and drug combinations.

Similar to PARP1 we could also observe PARP2 trapping (Supplementary Figure 6a), in line with previous biochemical data[22]. Analogous to PARP1, PARP2 trapping was cell cycle independent, yet the associated DNA damage signaling in the same cell population occurred primarily in S-phase cells (Supplementary Figure 6b). To our knowledge this is the first time that PARP1/2 trapping and DNA damage signaling can be directly correlated in the same cell population in a cell cycle resolved manner, and the results provide experimental support for previous models on PARPi-induced PARP trapping (across the cell cycle) and replication-associated formation of DNA lesions (in S-phase).

**Loss of PARP enzymes confers resistance to PARPi.** Acquired drug resistance in advanced disease is a major challenge for PARPi treatments and other targeted cancer therapies[5]. Mutation or down-regulation of PARP enzymes is one of the few mechanisms known to alleviate PARPi sensitivity and can lead to PARPi resistance[22,28,55,56]. PARP1 downregulation using siRNA indeed resulted in reduced S-phase-specific γH2AX signaling, providing further evidence that our read-outs reflect a cytotoxic consequence of PARPi-mediated PARP1 dysfunction (Supplementary Figure 7a). The rescue of DNA damage signaling was clearly detectable yet incomplete, which could be due to a contribution by other PARP family members. We therefore performed a series of PARP knockdowns and assessed DNA damage signaling upon exposure to three different PARPi (veliparib, olaparib, and talazoparib). Interestingly, while both PARP1 and PARP2 (but not PARP3) seemed to contribute to veliparib- and olaparib-induced γH2AX formation, talazoparib-induced DNA damage was markedly reduced only by PARP1 depletion (Supplementary Figure 7b,

c). Thus, the relative contribution of different PARP family members can be assessed for different PARPi in short-term phenotypic assays, which can complement fractionation-based trapping experiments in bulk populations[22,57].

Since poly(ADP-ribosyl)ation (PARylation) is controlled by both PARP activity and PAR degradation, we next asked whether deregulated PAR turnover would also impact PARPi-DDR. Interestingly, depletion of PARG, the main antagonizer of nuclear PARylation, resulted in reduced levels of S-phase-specific DNA damage signaling upon PARPi exposure, indicating that PARG-mediated turnover of PAR may contribute to PARPi toxicity (Supplementary Figure 8a). While we did not observe extenuated PARP1 or PARP2 trapping upon PARG knockdown (Supplementary Figure 8b, c), depletion of PARG did rescue PAR levels in PARPi-treated cells (Supplementary Figure 8d), suggesting that the knockdown was functional and that loss of PARG can impact the efficiency with which PARPi block cellular PARylation. Moreover, we observed that the PARPi-induced formation of RAD51 and BRCA1 foci was eased in PARG-depleted cells, indicative of partially restored PARP functions under these conditions (Supplementary Figure 9a, b). While elucidating the exact mechanism of relieved PARPi toxicity upon PARG loss needs further work, these results suggest that PARG expression and its activity may be relevant parameters of PARPi toxicity and resistance.

**Multidimensional analysis of PARPi-induced cell cycle arrest.** As additional parameters of cytotoxicity we assessed the consequences of PARPi-DDR on cell cycle progression. Short-term experiments confirmed that PARPi-induced DNA damage signaling is detectable within 30 min of inhibitor exposure (Fig. 5a) and found that the highly potent PARPi talazoparib induces higher amounts of S-phase damage compared to olaparib (Fig. 5a). When cells were incubated for 24 h with increasing inhibitor concentrations and stained for Cyclin A to allow for two-dimensional cell cycle staging, we observed a pronounced and dose-dependent accumulation of cells in S/G2 and, consistent with the degree of DNA damage induction, the S/G2 accumulation was stronger in talazoparib as compared to olaparib treated cells (Fig. 5b, c). For a more detailed analysis we performed 4-dimensional (4D) cell cycle staging based on DAPI, Cyclin A, EdU and the mitotic marker H3pS10 (Serine10-phosphorylated histone H3) (Supplementary Figure 10a–c). Whereas olaparib-treated cells slowed down DNA replication and accumulated in S/G2, talazoparib-treated cells had even more severe problems during S-phase progression and accumulated mostly there (Supplementary Figure 10a, b). Both compounds resulted in loss of mitotic cells marked by H3pS10, and, consistent with the effects on S-phase progression, this was more pronounced for talazoparib as compared to olaparib (Supplementary Figure 10c). Thus, PARPi-DDR measurements can be flexibly adjusted to monitor and evaluate consequences of PARPi at multiple cellular levels.

**Exploring drug interactions by quantitative imaging.** PARPi synergize with checkpoint kinase inhibitors such as inhibitors of the DNA damage response kinase ATR (ATRi)[58,59], however the underlying mechanisms are incompletely understood. To investigate this synthetic lethal interaction in more detail, we plotted total versus mean DAPI intensities of olaparib-, ATRi- and olaparib/ATRi-treated U-2 OS cells and assessed γH2AX formation. γH2AX signaling occurred primarily in cells with high mean DAPI intensities, i.e., cells with condensed chromatin (Fig. 6a). Co-staining with H3pS10 validated that cells with high mean DAPI intensities were mitotic cells and showed that ATRi treatment increased the percentage of PARPi-exposed cells

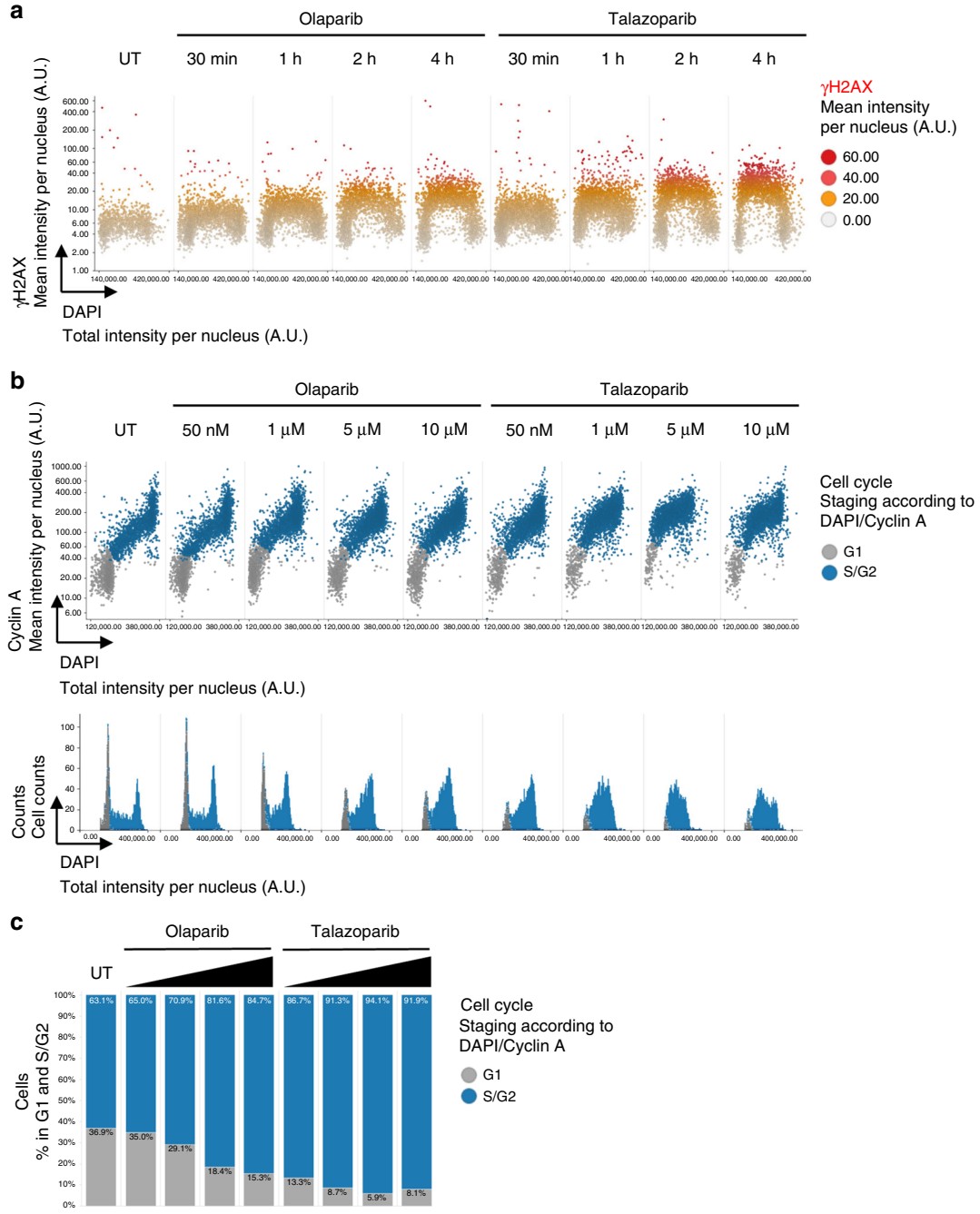

**Fig. 5** PARPi-induced cell cycle arrest revealed by quantitative imaging. **a** U-2 OS cells were treated with PARPi olaparib or talazoparib (both 10 μM) as indicated and stained for γH2AX and DNA content. Cell cycle resolved γH2AX profiles are shown. **b** Cells were treated for 24 h with different concentrations of PARPi olaparib or talazoparib as indicated and stained for Cyclin A and DAPI. Two-dimensional cell cycle profiles based on Cyclin A and DAPI intensities per nucleus are shown. One-dimensional cell cycle profiles are shown below. **c** Percentages of cells in G1 and S/G2 respectively. Color codes as defined in the figure panels

progressing into mitosis, where they experience high levels of DNA damage (Fig. 6b and Supplementary Figure 11a–d). To further exploit the predictive power of cell cycle resolved feature extraction from asynchronous cell populations we measured chromosome condensation and chromosome area in mitotic cells. This revealed that upon combined PARPi/ATRi treatment mitotic chromosomes were overall more condensed (Fig. 6c) and occupied a smaller area (Fig. 6d). Metaphase spreads were in line with this observation and showed significant chromosome shattering in the combined treatment (Fig. 6e). To consolidate this result, we performed live cell imaging experiments for up to 48 h

covering two rounds of cell division in control cells using H2B-GFP U-2 OS cells (Supplementary Figure 12a). ATRi-treated cells showed an extended mitotic duration, which was associated with lagging chromosomes, anaphase bridges and micronuclei formation upon completion of mitosis (Supplementary Figure 12b). PARPi treatment resulted in a G2 arrest (Supplementary Figure 12c), consistent with our previous results (Fig. 5 and Supplementary Figure 10). Combined PARPi/ATRi treatment, however, resulted in severely extended mitotic duration, eventually leading to catastrophic damage and genome disintegration (Supplementary Figure 12d). Consistently, CDK inhibition to

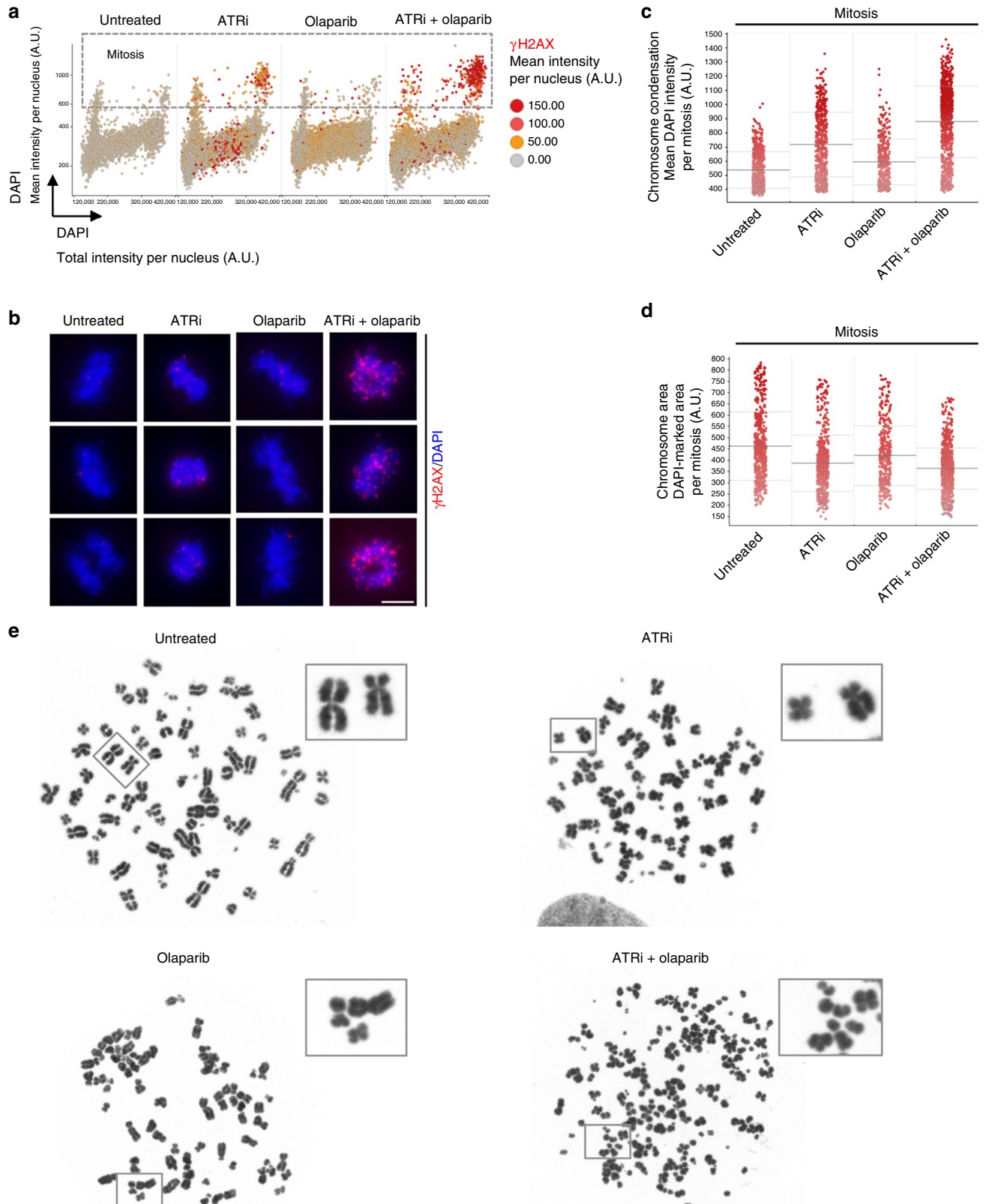

**Fig. 6** Synthetic lethality between PARPi and ATRi. **a** U-2 OS cells were treated for 24 h with olaparib (10 μM) and ATRi (1 μM) as indicated. Total versus mean DAPI intensities color-coded for γH2AX are shown. Mitotic cells are characterized by 4C (prophase to metaphase) or 2C (anaphase to telophase) DNA content and an increased mean DAPI intensity. **b** Representative images of mitotic cells after single and combined treatment. **c** Mean DAPI intensities of mitotic cells as a measure of chromosome condensation are shown. Horizontal lines represent population averages ± s.d. **d** Chromosome area of mitotic cells. Horizontal lines represent population averages ± s.d. **e** Metaphase spreads of cells treated as indicated. Color codes as defined in the figure panels. Scale bars, 10 μm

block mitotic entry abolished chromosome breakage and γH2AX formation in PARPi/ATRi-treated cells (Supplementary Figure 13a).

Replication intermediates occurring upon replication stress frequently escape cell cycle checkpoints and can be transmitted to the next cell cycle, where they are marked by the genome caretaker protein 53BP1[60]. We found 53BP1 nuclear bodies to increase drastically upon combined PARPi/ATRi treatment, suggesting that those cells, which can exit mitosis, do so with elevated inherited DNA damage (Supplementary Figure 11e). Besides its function in controlling the G2/M transition, ATR promotes RAD51 recruitment to stalled or collapsed replication forks and fosters DNA repair by homologous recombination[61]. Consistently, ATRi suppressed olaparib-induced RAD51 foci formation in S-phase cells (Supplementary Figure 13b), indicating that ATRi and PARPi synergize during S-phase progression, and that S-phase-born DNA lesions are then transmitted to mitosis, where, depending on the severity of the damage, they can either cause catastrophic chromosomal shattering or give rise to greatly elevated inherited DNA damage in the following G1 phase.

Last, to evaluate whether the approach could help uncover novel mechanisms of drug synergism beyond PARPi/ATRi, we performed a targeted siRNA-based pilot screen focusing on a small set of custom-selected cell cycle checkpoint and ATR/CHK1-related genes. We transfected cells in a 96-well format, left one set untreated and treated a second set for 8 h with olaparib, and measured DAPI, EdU and γH2AX intensities (Fig. 7a). Using three independent siRNAs per gene we analyzed around 2 million parameters of more than 200,000 cells. When we ranked genes according to γH2AX levels in EdU-positive S-phase cells we found siPARP1 to have the most negative z-score (Fig. 7b), consistent with our previous results on alleviated PARPi toxicity upon loss of PARP1. On the other extreme, we identified the replication checkpoint mediators RAD9A and NEK8[61,62], as well as TIME-LESS, which, in addition to promoting replication fork stability, was recently shown to interact with PARP1 and foster DSB repair[63,64]. We also identified SKP1 and CUL1, two components of the SKP, Cullin, F-box containing complex (SCF complex), as regulators of PARPi toxicity. This caught our attention, as SCF inhibition is currently assessed in phase I/II clinical trials to treat malignancies[65]. Re-investigation of our screening data confirmed that all three CUL1 and SKP1 siRNAs sensitized to olaparib (Fig. 7c and data not shown). SCF targeting in clinical tests is achieved by the NEDD8 inhibitor pevonedistat (MLN4924)[66], and we therefore wondered whether pevonedistat would synergize with PARPi. Long-term clonogenic survival assays showed that a combination of pevonedistat and olaparib was significantly more cytotoxic than single drug treatments (Fig. 7d). While PARP1 levels were not altered upon pevonedistat treatment (Supplementary Figure 14a–c), we observed elevated PARP1 trapping when pevonedistat was combined with olaparib (Fig. 7e). Normal formation of PARPi-induced RAD51 and BRCA1 foci suggested that pevonedistat-treated cells are not compromised in BRCA1/2 functions (Supplementary Figure 14d, e). The elevated PARP1 trapping, however, prompted us to test whether the synergism between pevonedistat and olaparib required PARP1 itself. Remarkably, down-regulation of PARP1 alleviated both the DNA damage signaling and the cell cycle arrest induced by the pevonedistat/olaparib combination (Fig. 7f). Thus, NEDD8/SCF inhibition by pevonedistat causes a hitherto uncharacterized cytotoxic interaction with PARPi, which depends on the presence of PARP1. Although the exact mechanism of interaction awaits further studies, our data suggest that it works at least partially via PARP1 trapping, raising the possibility that cancer cells with high expression of PARP1, even in a BRCA1/BRCA2-proficient scenario, may be particularly vulnerable to combined pevonedistat/PARPi treatments.

## Discussion

PARPi represent the first class of targeted therapeutics, which have been approved for cancer treatment based on the concept of synthetic lethality. While the molecular targets of PARPi are known and their biochemical activities have been characterized extensively, the cellular responses to PARPi have remained largely elusive. Defining the road from inhibiting PARP enzymatic activity to PARPi-induced cell death will be crucial to tackle the "holy trinity" in personalized cancer therapy[28]: deciding whom to treat based on biomarker-guided patient stratification, combating drug resistance by identifying mechanisms and predictive markers of reduced PARPi sensitivity, and optimizing combination therapy. Here we report an easy-to-implement and cost-efficient experimental pipeline that monitors, at the single-cell level and with sub-cellular resolution, the dynamics of the cellular response to PARPi in a sensitive and reliable manner, and can thereby aid all three areas of PARPi-based cancer therapy (Fig. 8).

Employing multidimensional high-content microscopy we present insights into cytotoxic interactions between PARPi and other clinically relevant agents (e.g., ATRi, topoisomerase blockers, and environmental toxins such as formaldehyde), and provide detailed views on the cellular responses to PARPi, their progression over time, and how deregulation of PARP enzymes impacts PARPi-induced DNA damage signaling. By simultaneously assessing PARPi-induced PARP trapping and γH2AX induction, we show that although PARP trapping occurs in all phases of the cell cycle, the DNA damage signal primarily occurs in S-phase cells. These results support the notion that cytotoxic PARPi-provoked DNA lesions arise in the context of DNA replication. The type(s) of lesions underlying PARPi toxicity potentially comprise multiple deleterious structures, including DNA-protein complexes, which form upon PARP enzymes being locked in an inactive conformation and which may pose an obstacle for the replication machinery, replication-associated conversion of DNA single- into double-strand breaks, reversed and degraded replication forks, as well as hitherto uncharacterized cytotoxic PARPi-induced structures. While complementary experimental approaches are needed to investigate the exact nature of these structures, QIBC-guided analyses are powerful tools for phenotypic explorations of drug interactions. Our data suggest that pevonedistat, a novel pharmacologic inhibitor of the NEDD8/SCF system currently being tested in phase I/II clinical trials[66], sensitizes cells to PARPi. Importantly, our findings further suggest that the synergism between PARPi and pevonedistat may rely on elevated levels of DNA lesions rather than compromised HR function. Interestingly, we show a PARP1 dependency and increased PARP1 trapping upon combined PARPi/pevonedistat treatment, suggesting that the NEDD8/SCF machinery may play a role in regulating PARP1 function and chromatin retention. These findings provide a rationale for future studies to mechanistically dissect the interplay between the SCF complex and PARP1 in the context of PARPi and to assess its clinical potential.

We envision that PARPi-DDR measurements as presented here can complement existing cell population-based proteomics and genomics approaches to interrogate physiological PARP functions[67–69], and that it will prove valuable in basic and pre-clinical research to predict, identify and characterize mechanisms of PARPi sensitivity and resistance and explore in targeted assays and large scale combinatorial screens new drug combinations for synergistic effects. Provided that suitable markers can be generated and smartly combined in cells to assess PARPi effects live (e.g., antibody-independent surrogate markers for DNA damage signaling, HR function, cell cycle progression, etc.) high-content time-lapse imaging may provide even more detailed views on drug cytotoxicity in the future. Given the general adaptability of image-based single-cell assays to pre-clinical and clinical research

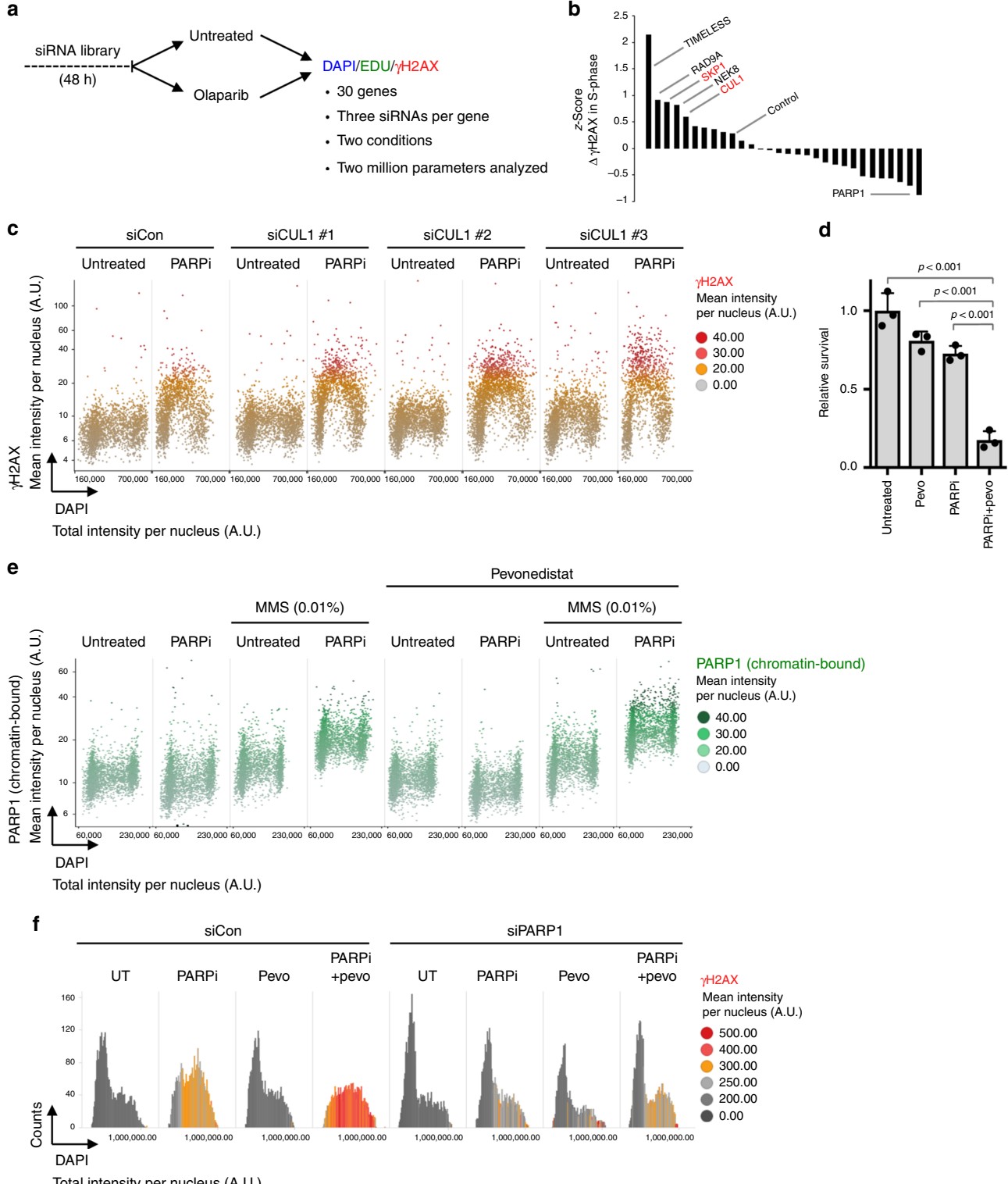

**Fig. 7** A functional interaction between inhibition of NEDD8/SCF and PARPi. **a** Overview of the siRNA screening procedure. For each well, EdU incorporation, DNA content and γH2AX signaling was quantified at the single cell level. **b** z-score according to γH2AX in S-phase cells. **c** Cell cycle resolved γH2AX profiles obtained from the screen for cells transfected with 3 siRNAs for CUL1 are shown. **d** U-2 OS cells were treated with 1 μM olaparib and 10 nM pevonedistat (pevo), and cell proliferation was evaluated by clonogenic survival. Data are presented as mean ± s.d. (n = 3). P-values were obtained by unpaired t-test. **e** U-2 OS cells were treated with 10 μM olaparib and 0.01% MMS for 4 h in the presence or absence of pevonedistat (100 nM, added 4 h prior to and kept on during the PARPi/MMS treatment), pre-extracted on ice in 0.2% Triton X-100 for 2 min to remove soluble, non-chromatin-bound proteins, and stained for PARP1 and DNA content. Chromatin-bound PARP1 levels were quantified and are depicted as a function of cell cycle progression. **f** U-2 OS cells were transfected with siRNA against PARP1, treated with 10 μM olaparib and 100 nM pevonedistat for 24 h, and stained for γH2AX and DNA content. Cell cycle profiles are shown. Color codes as defined in the figure panels

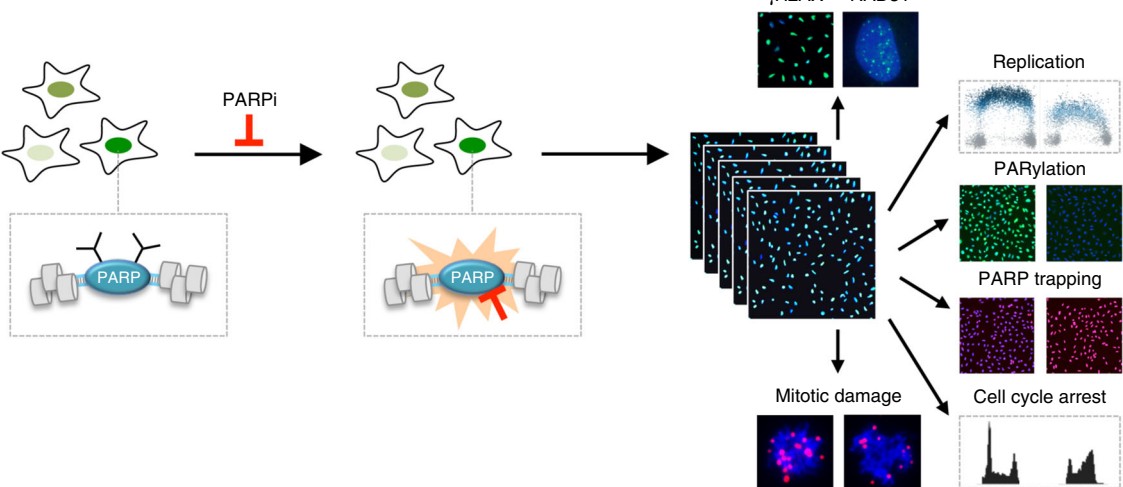

**Fig. 8** Multidimensional analysis of PARPi toxicity by cell cycle resolved automated high-content microscopy. Key parameters of the cellular response to PARPi are simultaneously quantified, including DNA damage signaling (measured by γH2AX formation), activation of the homologous recombination repair pathway (measured by RAD51 foci formation), slow-down of DNA replication (measured by EdU incorporation), inhibition of PARP enzymes (measured by changes in poly(ADP-ribose) (PAR) levels), trapping of PARPs (measured by PARP levels after in situ fractionation), cell cycle arrest (measured by accumulation of cells in specific cell cycle phases), and mitotic chromosomal damage (measured by γH2AX on mitotic chromosomes). The experimental pipeline is equally suited for individual assays and large scale high-throughput screens

settings such developments may eventually help predict clinical outcomes and stratify cancer patients.

## Methods

**Cell culture and drug treatments**. All cells were grown in a sterile cell culture environment and routinely tested for mycoplasma contamination. Human U-2 OS cells (authenticated by STR profiling), U-2 OS derived H2B-GFP cells and hTERT-RPE1 cells (ATCC), were grown under standard cell culture conditions (humidified atmosphere, 5% $CO_2$) in Dulbecco's modified Eagle's medium (DMEM) containing 10% fetal bovine serum (GIBCO) and penicillin-streptomycin antibiotics. MDA-MB-436 cells were provided by Mohamed Bentires-Alj, University of Basel, and were grown in DMEM containing 10% fetal bovine serum and penicillin-streptomycin antibiotics. HCC1143 cells were also provided by Mohamed Bentires-Alj, University of Basel, and were grown in RPMI-1640 medium containing 10% fetal bovine serum and penicillin-streptomycin antibiotics. For pulsed EdU (5-ethynyl-2′-desoxyuridine) (Thermo Fisher Scientific) incorporation, cells were incubated for 20 min in medium containing 10 μM EdU. The Click-iT EdU Alexa Fluor Imaging Kit (Thermo Fisher Scientific) was used for EdU detection. Unless stated otherwise the following compounds were used in this manuscript at the indicated final concentrations: Az-20 (1 μM) (Tocris), Camptothecin (50 nM to 1 μM) (Selleckchem), Temozolamide (1 mM) (T2577, Sigma), Roscovitine (20 μM) (Selleckchem), PJ-34 (10 μM) (ALX-270–289-M001, Enzo Life Sciences), niraparib (MK-4827) (10 μM) (S2741, Selleckchem), rucaparib (AG-014699, PF-01367338) (10 μM) (S1098, Selleckchem), talazoparib (BMN 673) (50 nM-10 μM) (S7048, Selleckchem), veliparib (ABT-888) (10 μM) (ALX-270–444-M005, Enzo Life Sciences), olaparib (AZD-2281) (50nM-10 μM) (S1060, Selleckchem), pevonedistat (MLN4924) (10 nM to 100 nM) (S7109, Selleckchem), Hydrogen peroxide (H3410, Sigma-Aldrich) (0.1 mM), Methyl methanesulfonate (129925, Sigma-Aldrich) (0.01%). X-ray irradiation of cells was performed with a Faxitron Cabinet X-ray System Model RX-650. For cell synchronization and release experiments, exponentially growing U-2 OS cells were incubated with 2 mM thymidine for 20 h and subsequently washed and cultured in fresh medium with or without olaparib (10 μM) for 6 h.

**siRNA transfections**. Duplex siRNA transfections were performed for 72 h with Ambion Silencer Select siRNAs using Lipofectamine RNAiMAX (Thermo Fisher Scientific). The following Silencer Select siRNAs were used at a final concentration of 25 nM: PARP1 (s1098), PARP2 (s19504), PARP3 (s19507), FANCD2 (s4988), BRCA1 (s459), BRCA2 (s2085). PARG knockdown (s16158) was performed for 36 h at a final concentration of 5 nM. When several siRNAs were combined, the final siRNA concentration was kept constant at 25 nM for all conditions. Negative control (s813) from Ambion was used as a non-targeting control and is abbreviated siCon. The siRNA-based screen was performed by reverse-transfection of U-2 OS cells cultured in CELLSTAR 96-well-plates (Greiner Bio-One) for 48 h at a cell density of 6000 cells per well at the time of transfection with Ambion Silencer Select siRNAs at a final concentration of 5 nM using HiPerFect (Qiagen) reagent.

**Histone extraction**. For acid extraction of histones, cell pellets were resuspended in Triton Extraction Buffer (TEB: PBS containing 0.5% Triton X 100 (v/v), 2 mM

phenylmethylsulfonyl fluoride (PMSF), 0.02% (w/v) $NaN_3$) at a cell density of $10^7$ cells per ml and lysed on ice for 10 min. Cells were centrifuged at 6500×$g$ for 10 min at 4 °C and then washed with TEB at a cell density of $2 \times 10^7$ cells per ml. Cell pellets were resuspended in 0.2 N HCl at a density of $4 \times 10^7$ nuclei per ml and extracted over night at 4 °C. Samples were centrifuged at 6500×$g$ for 10 min at 4 °C and the supernatants neutralized with 20% 1 M NaOH. Protein amounts were quantified using the standard Bradford method.

**Immunochemical methods**. Proteins were resolved by SDS-polyacrylamide gel electrophoresis (SDS-PAGE) and transferred onto polyvinylidene fluoride (PVDF) membranes. Membranes were blocked with PBS-Tween20 (0.01%) containing 5% milk powder for 1 h at room temperature. Primary antibodies in blocking solution were applied over night at 4 °C. The following primary antibodies were used for western blot analysis: Histone H3 (rabbit, Abcam ab1791, 1:20,000), H2AX Phospho S139 (mouse, Biolegend 613401, 1:500), PARP1 (rabbit, Santa Cruz sc-7150, 1:500), NFκB p65 (rabbit, Santa Cruz sc-109, 1:500), p27 (rabbit, Santa Cruz sc-528, 1:500), PCNA (mouse, Santa Cruz sc-56, 1:500). Secondary horseradish peroxidase-coupled antibodies (Vector labs VC-PI-1000-M001 and VC-PI-2000-M001, 1:10,000) were applied for 1 h at room temperature in PBS-Tween20 (0.01%) containing 1% milk powder prior to detection by ECL-based chemiluminescence. Western blot scans used to assembly the figures are provided in Supplementary Figure 15a–d.

**Immunostaining**. Cells were seeded onto sterile 12 mm glass coverslips inside 60 mm cell culture dishes, or into multiwell plates (CELLSTAR 96-well-plates, Greiner Bio-One), and grown for 24 h to reach a cell density of 70–90%. After applying the indicated cell treatments, cells were fixed in 3% formaldehyde in PBS for 15 min at room temperature, washed once in PBS, permeabilized for 5 min at room temperature in 0.2% Triton X-100 (Sigma-Aldrich) in PBS, washed twice in PBS and incubated in blocking solution (filtered DMEM containing 10% FBS and 0.02% Sodium Azide) for 15 min at room temperature. Where indicated, cells were pre-extracted in 0.2% Triton X-100 in PBS for 2 min on ice prior to formaldehyde fixation. All primary antibodies (see below for specifications) and secondary antibodies (Alexa Fluor 488 goat-anti rabbit, Alexa Fluor 488 goat-anti mouse, Alexa Fluor 568 goat-anti rabbit, Alexa Fluor 568 goat-anti mouse, Alexa Fluor 647 goat-anti rabbit, Alexa Fluor 647 goat-anti mouse, all Thermo Fisher Scientific, 1:500) were diluted in blocking solution. For antibody incubations the coverslips were inverted over 40 μl of the desired antibody for 1–2 h at room temperature. For immunostaining in multiwell plates, 40 μl of the desired antibody was used per well. When combining the staining with an EdU Click-it reaction, this reaction was performed prior to incubation with the primary antibodies according to the manufacturer's recommendations (Thermo Fisher Scientific). Following antibody incubations, coverslips were washed once with PBS and incubated for 10 min with PBS containing 4′,6-Diamidino-2-Phenylindole Dihydrochloride (DAPI, 0.5 μg/ml) at room temperature to stain DNA. Following three washing steps in PBS, coverslips were briefly washed with distilled water and mounted on 5 μl Mowiol-based mounting media (Mowiol 4.88 (Calbiochem) in Glycerol/TRIS). Multiwell plates were subjected to the same DAPI staining and washing procedure and were then

kept in PBS. The following primary antibodies were used for immunostaining: H2AX Phospho S139 (mouse, Biolegend 613401, 1:1000), RAD51 (rabbit, Bioacademia 70-002, 1:1000), BRCA1 (mouse, Santa Cruz sc-6954, 1:100), Poly(ADP-ribose) (rabbit, Enzo Lifesciences ALX-210–890, 1:1000), Cyclin A (rabbit, Santa Cruz sc-751, 1:100), PARP1 (rabbit, Santa Cruz sc-7150, 1:100), PARP2 (rabbit, Active Motif 39743, 1:250), Histone H3 phospho S10 (rabbit, Abcam ab5176, 1:2000), 53BP1 (rabbit, Santa Cruz sc-22760, 1:500).

**Quantitative image-based cytometry**. Automated multichannel wide-field microscopy for quantitative image-based cytometry (QIBC)[43–46] was performed on an Olympus ScanR Screening System equipped with an inverted motorized Olympus IX83 microscope, a motorized stage, IR-laser hardware autofocus, a fast emission filter wheel with single band emission filters, and a 12 bit digital monochrome Hamamatsu ORCA-FLASH 4.0 V2 sCMOS camera (2048 × 2048 pixel, 12 bit dynamics). For each condition, image information of large cohorts of cells (typically at least 500 cells for the UPLSAPO 40× objective (NA 0.9), at least 2000 cells for the UPLSAPO 20× objective (NA 0.75), and at least 5000 cells for the UPLSAPO 10× (NA 0.4) and UPLSAPO 4× (NA 0.16) objectives) was acquired under non-saturating conditions at a single autofocus-directed z-position. Identical settings were applied to all samples within one experiment. Images were analyzed with the inbuilt Olympus ScanR Image Analysis Software Version 2.5.1, a dynamic background correction was applied, nuclei segmentation was performed using an integrated intensity-based object detection module using the DAPI signal, and foci segmentation was performed using an integrated spot-detection module. All downstream analyses were focused on properly detected interphase nuclei or mitotic chromosomes containing a 2C-4C DNA content as measured by total and mean DAPI intensities. Fluorescence intensities were quantified and are depicted as arbitrary units. Color-coded scatter plots of asynchronous cell populations were generated with Spotfire data visualization software (TIBCO). Within one experiment, similar cell numbers were compared for the different conditions. For visualizing discrete data in scatter plots (e.g., foci numbers), mild jittering (random displacement of data points along the discrete data axes) was applied in order to demerge overlapping data points. Representative scatter plots and quantifications of independent experiments, typically containing several thousand cells each, are shown.

**Metaphase spreads**. U-2 OS cells were treated with olaparib (10 μM) and AZ-20 (1 μM) for 24 h, arrested with Colcemid (Roche) at a final concentration of 100 ng/ml during the last 12 h of the treatments, harvested by trypsinization, swollen in 75 mM KCl for 30 min at 37 °C and fixed in methanol:acetic acid 3:1. Aliquots of the cellular suspension were dropped onto microscopy slides to obtain chromosome spreads, which were stained with DAPI (0.5 μg/ml) and mounted with Mowiol-based mounting media. Image acquisition was performed on a Leica SP8 laser-scanning microscope equipped with solid-state diode lasers for 405 nm (50 mW), 488 nm (20 mW), 552 nm (20 mW), and 638 nm (30 mW) using an HCX PL APO CS2 63× immersion oil objective (NA 1.4).

**Time-lapse microscopy**. Time-lapse microscopy of H2B-GFP U-2 OS cells was performed on the Olympus ScanR Screening System under CO₂ (5%) and temperature (37 °C) control and employing an inbuilt infrared-based hardware autofocus. Exposure times were kept minimal to avoid phototoxicity. Cells were plated on multiwell plates (CELLSTAR 96-well-plates, Greiner Bio-One) at a density of 8000 cells per well 24 h prior to imaging. Images were taken at 15 min intervals for 48 or 72 h as indicated in the figure legends using a UPLSAPO 20× objective (NA 0.75). During the time-lapse imaging cells were cultured with FluoroBrite DMEM medium containing 10% FCS (GIBCO) and penicillin-streptomycin.

**Flow cytometry**. Cells were fixed with chilled 70% ethanol, permeabilized with 0.05% Tween-20 in PBS, incubated for 2 h at room temperature with an antibody recognizing H2AX Phospho S139 (mouse, Biolegend 613401, 1:500), followed by incubation with the secondary antibody against mouse immunoglobulin labeled with a green fluorophore (Alexa Fluor 488 Life Technologies, 1:500) during 1 h at room temperature. Finally, DNA was stained with propidium iodide (PI) (50 μg/ml). Samples were acquired and analyzed in a Fortessa LSRII flow cytometer (Beckton Dickinson). The generated data were processed using the Spotfire data visualization software (TIBCO).

**RNA extraction and quantitative PCR**. RNA was purified with TRIzol reagent (Life Technologies). RNA was primed with random hexamers (11034731001, Roche) and reverse-transcribed to cDNA using a MultiScribe Reverse Transcriptase (4311235, Thermo Fisher). qPCR was performed with the KAPA SYBR FAST qPCR Kit (KAPA Biosystems) on a Rotor Gene Q system (Qiagen). Relative transcription levels were obtained by normalization to Eif2c2 expression. The following primer pairs were used:

hPARP1 F:GCATCAGCACCAAAAAGGAGGTGG
hPARP1 R:GATTTGTTGATACCTTCCTCCTTGACCTGG
hPARP2 F:GTGGAGAAGGATGGTGAGAAAG
hPARP2 R: CTCAAGATTCCCACCCAGTTAC
hPARP3 F:GCAAGTCAGCTGGATATGTTATTG
hPARP3 R:CGTGTTGATATGGTGCTCTCT

hPARG F: CGAGCAGGAGAAGTTCCTAAAC
hPARG R: AGTTCGCTCACCATTCTCATC
hBRCA1 F: TGAAATCAGTTTGGATTCTGC
hBRCA1 R: CATGCAAGTTTGAAACAGAAC
hBRCA2 F: CCAAAGTTTGTGAAGGGTCG
hBRCA2 R: GTAGAACTAAGGGTGGGTGGTG
hFANCD2 F: AAAACGGGAGAGAGTCAGAATCA
hFANCD2 R: ACGCTCACAAGACAAAAGGCA
hEIF2C2 F:GTCCCTTTTGAGACGATCCAG
hEIF2C2 R:AGCCAAACCACACTTCTCG

**Chromatin fractionation**. Cell pellets were collected by scraping in 1× PBS with 1× protease inhibitor (cOmplete, Roche) and split in two equal fractions: (A) For the total proteome cells were resuspended in 1× MNase buffer (0.3 M Sucrose, 50 mM Tris pH 7.5, 30 mM KCl, 7.5 mM NaCl, 4 mM MgCl₂, 1 mM CaCl₂, 0.125% NP-0.4, 0.25% Na-Deoxycholate) with 1× protease inhibitor and 10 U of MNase (10107921001, Roche) for every 5 million cells. Cells were incubated for 30 min at 37 °C, boiled in 1× SDS-loading buffer for 5 min, spun down at 16,000×g for 5 min and the supernatant was collected for western blot. (B) For chromatin-bound and soluble fractions cells were resuspended in chromatin extraction buffer (10 mM Hepes pH 7.6, 3 mM MgCl₂, 0.5% Triton-X-100, 1 mM DTT) with 1× protease inhibitor. Cells were rotated for 30 min at room temperature and spun down at 1300×g for 10 min at 4 °C. The supernatant was collected, spun down at 16,000×g for 5 min at 4 °C and the supernatant was collected again (soluble fraction), boiled in 1× SDS-loading buffer for 5 min, spun down at 16,000×g for 5 min and the supernatant was collected for western blot. The pellet of the chromatin extraction step (chromatin-bound fraction) was resuspended in MNase buffer with 1× protease inhibitor and 10U of MNase for every 5 million cells. Cells were incubated for 30 min at 37 °C, boiled in 1× SDS-loading buffer for 5 min, spun down at 16,000×g for 5 min and the supernatant was collected for western blot. Protein amounts were quantified using the standard Bradford method before addition of SDS-loading buffer.

**Clonogenic survival assay**. U-2 OS cells were seeded at single cell density and 24 h later incubated with olaparib (1 μM), pevonedistat (10 nM) or a combination of both. Cells were incubated for 10 days and the number of colonies with more than 50 cells was counted after staining with crystal violet (0.5% crystal violet in 20% ethanol).

**Statistical analysis**. Mann–Whitney test was used for statistical analysis of microscopy-derived intensity data. Unpaired t-test was used for clonogenic survival data.

**Data availability**. All relevant data are available from the authors.

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

## Acknowledgements

We acknowledge the Flow Cytometry Facility and the Center for Microscopy and Image Analysis at the University of Zurich, in particular Urs Ziegler and José María Mateos Melero, for excellent support, Luis Toledo for H2B-GFP U-2 OS cells, Mohamed Bentires-Alj for MDA-MB-436 and HCC1143 cells, and Michael Hottiger for chemical compounds. We thank all members of the Altmeyer lab for valuable discussions and experimental help. Research in the lab of Matthias Altmeyer is supported by the Swiss National Science Foundation (SNSF Professorship Grant PP00P3_150690 and PP00P3_179057), the European Research Council (ERC) under the European Union's Horizon 2020 research and innovation program (ERC-2016-STG 714326), the Novartis

Foundation for Medical-Biological Research (Grant 16B078), and the Swiss Foundation to Combat Cancer (Stiftung zur Krebsbekämpfung). J.M. is supported by the Gobierno Vasco Programa Posdoctoral de Perfeccionamiento de Personal Investigador Doctor. A. L. is a member of the Cancer Biology Program of the Life Science Zurich Graduate School. F.T. is a member of the Molecular Life Sciences Program.

## Author contributions

J.M., A.L., F.T., T.S. and R.I. conducted experiments. All authors analyzed, interpreted and visualized data. M.A. conceived the study, supervised the work and wrote the original manuscript draft. All authors reviewed and edited the manuscript.

## Additional information

**Competing interests:** The authors declare no competing interests.

