## [Peer Review File · Nature Communications]

Reviewers' comments:

Reviewer #1 (Remarks to the Author):

In their manuscript entitled: 'Quantitative analysis of PARP inhibitor toxicity by multidimensional fluorescence microscopy reveals mechanisms of sensitivity and resistance', Michelena and co-workers used high-throughput microscopy to analyse cellular effects of PARP inhibition.

Overall, I am impressed by the quality of the data obtained by QIBC analysis. The sensitivity indeed is better when compared for instance to flow cytometry analysis. I am also sure that using this technique biological mechanisms can be uncovered that cannot be solved using less advanced techniques.

However, to be honest, in this manuscript, I have not seen cellular phenotypes upon PARP inhibitor treatment, that were not observed previously, or could not be done using more conventional analyses. We already knew that PARP inhibition requires its trapping activity to induce cytotoxicity (Murai, Pommier), and that PARP inhibition requires S-phase to induce DNA lesions. Also the recruitment of Rad51 (dependent on BRCA1/2) is in line with previous findings, and the sensitizing effects of ATR inhibition to PARP inhibition have been described previously. The aspects that are relatively new are the role of PARG and the effects of individual PARP isoforms, although in depth mechanistic insight for these observations is not provided. Further, the manuscript lacks technical details and proper controls, as indicated below.

Overall, to my opinion the current manuscript lack sufficient novelty and impact to warrant publication.

Specific Points:

In the abstract, a few claims are made that are not substantiated by the results: 'our approach can be adapted to predict outcomes' and '...ultimately stratify cancer patients'. If anywhere, these statements should be in the discussion.

If the authors want to make a statement that QIBC is required to predict outcome to a range of parp inhibitors at early time-points, then a comparison to an alternative technique is required. Previously, the various PARP inhibitors were already compared by the Pommier lab, in assays that only required few hours of PARP inhibitor treatment, which also predicted cytotoxicity of the compounds.

The hypercondensation that is observed upon combined treatment with olaparib and ATRi may be due to prolonged time in mitosis. To control for this, the authors include MG-132 treatment. However, mitotic hypercondensation is time-dependent, and so the time of treatment should be taken into account. In this case, the MG-132 treatment was done much shorter than the other treatments. For a fair comparison, the time that cells spend in mitosis should be taken into consideration.

Related to the point above, the NCAPH2 localization to mitotic chromosomes needs to be performed in all treatments, to see if this is lost specifically in the combined treatments. Also, these results should be quantified from multiple cells, and statistics should be applied. Regardless, this might be an epiphenomenon.

The observation that combined PARPi and ATRi treatment leads to increased cytotoxicity is stated to 'likely' be due to mitotic aberrancies. Firstly, it is not clear in which cell type these experiments were performed. If performed in a HR-proficient cell line, ATR inhibition may induce HR deficiency which could underlie the observed effects. Also, the current experiments do not point at which of the many functions of ATR underlie the observed effects. Loss of cell cycle checkpoint behaviour? DNA repair? Fork stability?

-knockdown should be verified when used in experiments (Fig 2/5). WB or qPCR data should be included.

-it is unclear from the figure legends or figures which cells are used, and which concentrations of drugs were applied.

-Figure 6: mitotic duration should be quantified. Please quantify G2 arrest. Please indicate G2 arrest separately from S/G2 arrest.

Minor points:

-The U2-OS cell lines does not strike me as a very logical cell line to uncover PARPi effects, since the authors start the paper to identify how PARPi actually cause cytotoxicity, because this may guide treatment decisions and patient selection. A more logical choice would be ovarian/breast cancer models with/without HR defects.

-Lynparza is a brand name and is spelled with a capital, all the PARP inhibitor names (olaparib, talazoparib etc) are spelled with small fonts.

-Page 5: 'and quantified all relevant parameters of PARPi toxicity'. This is a strong overstatement. There are multiple parameters of PARPi toxicity that may be important for PARPi toxicity (eg P-glycoprotein expression as the authors earlier indicate).

-In the introduction, it is mentioned that Lynparza is approved, and other PARP inhibitors are in late clinical testing. Actually, Rubraca and Zejula have also been approved.

Reviewer #2 (Remarks to the Author):

Here J. Michelena et al. develop a sensitive single cell assay to assess cell cycle specific responses to PARPi after relatively short term treatments. This is complementary (and perhaps more sensitive and quantitative) than current protocols which measure population averages in long-term clonogenic survival experiments. A few nice examples of the utility of the method are given, including how ATRi synergizes with PARPi, measurements of PARPi induced PARP1 trapping, as well as the demonstration that PARG-mediated turnover of PAR contributes to PARPi toxicity. This is a solid report which will be useful for the scientific community.

They should address the following questions prior to publication:

- 1) They demonstrate that defects in HR sensitize to PARPi. They note differences in gamma H2AX at 16 and 48 hours post-Parpi treatment (Figures 2a-c). Why don't they see differences earlier- after all, in Figure 1 they see a PARPi responses in S phase already by 15 minutes. If they don't see a difference in DDR signaling between WT and HR deficient early on, they should explicitly show and explain this.
- 2) PARPi and formaldehyde toxicity synergize- Figure extended Data Fig. 3b. Is this only in BRCA2 deficient cells, or also WT or BRCA1 deficient?
- 3) How does depletion of PARG (which increases PAR levels) and depletion of PARP1/PARP2 (which reduces PAR levels) both result in reduced levels of DDR signaling? This is not intuitively obvious. Does reduction of PARG reduce PARPi induced trapping? Does PARG depletion also result in cisplatin resistance?
- 4) Combined PARPi/ATRi lead to gamma H2AX specifically in mitotic cells. This was examined visually by DAPI staining (Figure 7). They should also quantify by QIBC using a mitotic marker

(H3S10p and MPM2-p).

Reviewer #3 (Remarks to the Author):

PARP inhibitors (PARPi) have been utilized in the clinical treatment for Stage IV ovarian cancers with BRCA mutations. In this study, the authors used fluorescence microscopy and quantitative image-based cytometry to examine the defects of DNA damage response induced by PARPi treatment. Although this novel approach may be useful for analyzing the efficacy of PARPi, the study lacks mechanistic insight. Furthermore, the overall quality of the study does not reach the level of Nature Communications.

Specific comments:

1. In the study, the authors show that PARPi treatment causes DNA damage in the S phase. However, it is unclear how PARPi induces DNA damage exclusively in the S phase, and what type of lesions is caused by PARPi treatment.
2. Fig. 1b lacks a mock treatment control. A large field of images with gH2AX positive cells should be shown.
3. In Fig. S1d, the authors did not find obvious gH2AX using Western blot in unsynchronized cells. However, the authors should harvest the S phase cells and analyze gH2AX in the S phase population.
4. Statistical analysis is lacking in Fig. S1e. It is unclear if ola treatment has any significant effect.
5. In Fig. 1C, the authors used 10 uM of 5 different PARPis to examine gH2AX. However, the IC50s of these PARP1 are generally below 1 uM. And the IC50 is different in different cancer cell lines. Thus, the conclusions are invalid.
6. In Fig. S2, the authors treated cells with ola for 8 hours and observed obvious RAD51 foci in the S phase cells. It indicates that numerous DSBs have been generated. However, it is unclear how cells were still evenly distributed in different cell cycle phases and did not activate cell cycle checkpoints.
7. In Fig. 2B, in the BRCA1 or BRCA1 knockdown cells, PARPi treatment induced cell accumulation in the G1 and G2 phases. What is the underlying mechanism? In Fig. 2C, When FNACD2 was down-regulated, cells were only arrested in G2/M with PARPi treatment. Thus, the underlying mechanism could be totally different from those in the BRCA1 or BRCA1 knockdown cells.
8. In Fig. 3a, the chromatin-bound PARP1 should be validated with Western blot. In addition, the chromatin-trapped PARP1 is not correlated with gH2AX because gH2AX is only elevated in the S phase cells.
9. Fig. 4c is more problematic. The second sample should be same as the third sample in Fig. 3b. However, there are totally two different pattern of gH2AX.
10. In Fig. 5b, with ola treatment, PARP1/2 were still trapped in DNA lesions. How can knockdown of PARG suppress gH2AX. What is the underlying mechanism?
11. Ola does not suppress or trap PARP3. How can PARP3 knockdown rescue the gH2AX phenotype induced by ola treatment in Fig. S5c.
12. Data in Fig.7 are irrelevant with other data in the rest of the manuscript.

Reviewer #4 (Remarks to the Author):

The authors present a well written manuscript with an evident logic progression. The presented method, Quantitative Image Based Cytometry (QIBC), allows for the analysis of PARPi toxicity, on sub-population resolution, by the use of microscopy based single-cell analyses combined with cell cycle staging of an unsynchronized cell population. The applicability of QIBC for the comparison of individual PARP inhibitors and combination treatments in a cell cycle resolved manner is extensively demonstrated.

Furthermore, the authors elegantly exploit the possibilities of sub-cellular analysis, provided by microscopy based methods in contrast to flow cytometry, by extending their analysis from nuclear average values to the detection of nuclear RAD51 foci.

The presented method has been validated against state of the art methods by including the analysis of PARP inhibitor toxicity, PARP enzyme trapping and DNA damage response. A clear advantage in sensitivity and required treatment duration has been demonstrated and is attributed to the technological and methodological improvement achieved by single cell analysis, cell cycle staging and high throughput microscopy.

The manuscript is, however, a bit of a chimera, one part is the presentation of a new method and the other part is trying to elucidate PARP biology in cell cycle, disease and therapy. Both parts are interesting, but none is totally convincing:

- 1) The new QIBC method is not as novel as claimed. At the end of the day, the authors describe the cellular response to PARP inhibition in combination with siRNA mediated manipulation of a test cell line. The efficiency and specificity of their siRNA treatments is not shown. The cellular response is determined mostly pairwise, which is OK, but not novel. Here real multi-parameter analysis would have been better and perhaps even revealing.
- 2) In their abstract and introduction, the authors paint a rather dire picture of the state of the art and neglect far more advanced HTS/HCA studies. This being mostly a new methods manuscript, the authors should spend more energy describing prior work, e.g., when it comes to cell cycle staging the authors might want to mention similar protocols described before (e.g. the "Cell cycle staging of individual cells by fluorescence microscopy" Roukos et al., Nat Protoc. 2015 Feb;10(2):334-48). Currently, most of their citation are either PARP or their own.
- 3) The last sentence of the abstract: "Our approach can be adapted to predict outcomes of a variety of cytotoxic agents and ultimately stratify cancer patients." promises far more than the manuscript actually delivers and is confusing not to say misleading. This is the underlying but never substantiated claim throughout this manuscript. The test of other cytotoxic agents other than PARP inhibitors and patient derived cells with known lesions would have been necessary and helpful.
- 4) Most measurements were performed with fixed and stained cells. Seeding "cells onto clean and sterilized glass coverslips", is rather low throughput and a bit outdated, certainly not suited for automated medium to high throughput screenings (HTS) and live cell readouts would have likely been more informative. Having not done so, certainly would have deserved some discussion.
- 5) Having produced large data sets, the authors could and should have taken advantage of this resource and done some rigorous statistical testing/analyses. How robust is the method and when is a difference a significant difference?
- 6) The coverage of prior publications on PARP biology is far from satisfying, as many insights into PARP function, interactions with cellular pathways and the effect of inhibitors are not cited/discussed. All conclusion about the role of PARPs, PARG, the cell cycle and homologous recombination are based on simple correlations and would either have to be discussed with much greater caution or studied much more thoroughly with systematic approaches and multiple complementing lines of evidence.
- 7) Did the authors evaluate the effects of PARPi treatment on non-responsive cell lines?
- 8) Is the method transferable to multiwell format? This would greatly improve throughput and screening capability. Especially in the context of predictive biomarker and drug screening on patient samples as well as the screening of large compound libraries in minimal volumes would be of great benefit.
- 9) Could the authors please clarify whether images of individual confocal planes or z-projections over multiple planes were used for quantification?
- 10) Finally, the authors might want to revise sloppy jargon like e.g. at the end of the introduction: "by allowing replication-born lesions to prematurely enter mitosis" - lesions do not enter mitosis, but cells might.

Point-by-point response to the reviewers

We would like to thank all four reviewers for having taken the time to read and critically evaluate our work. The comments were very valuable and helped us to carefully revise and extend our technology manuscript. We were particularly excited to read that the reviewers were 'impressed by the quality of the data' (reviewer 1), praised the clear advantage in sensitivity and quantitative power of the presented technique (reviewers 1, 2 and 4), acknowledged the extensively demonstrated applicability/utility (reviewers 2 and 4), and considered the approach 'novel' (reviewer 3) and our manuscript 'solid' and 'useful for the scientific community' (reviewer 2). All this was very encouraging, and we indeed hope to present the benefits of a microscopy-based single cell technique, which is currently used only by a handful of research groups worldwide, to the scientific community with the vision that it has broad applicability and can significantly aid basic research, be used in drug discovery screens and in screens for synthetic lethal interactions, and may eventually be used to help stratify cancer patients.

One of the major advancements in our revised manuscript, which was inspired by the constructive suggestions of the reviewers, is represented by targeted high-content imaging screens, which, using multidimensional read-outs with single cell resolution, allowed us to identify a hitherto uncharacterized synthetic lethal interaction between PARPi and pevonedistat, a drug that is currently used in clinical trials to inhibit the SCF machinery and treat patients suffering from myeloma, lymphoma, metastatic melanoma and other advanced solid tumors. Although the exact molecular mechanism of its interaction with PARPi remains to be characterized in future studies, we show in our revised manuscript that it is linked to elevated PARP1 trapping and that it requires the presence of PARP1 to be effective. We are very excited about these new additions, which we present in new Figure 7 and Extended Data Figure 14, and we are grateful to the reviewers for having inspired us to perform these experiments. In addition, and again based on the reviewers' constructive comments, we included important controls and performed validation experiments, which resulted in 40 new figure panels, incorporated into a total of 8 Main and 14 Extended Data Figures. All new additions and changes to the manuscript text are explained in our point-by-point response below:

Reviewers' comments:

Reviewer #1 (Remarks to the Author):

In their manuscript entitled: 'Quantitative analysis of PARP inhibitor toxicity by multidimensional fluorescence microscopy reveals mechanisms of sensitivity and resistance', Michelena and co-workers used high-throughput microscopy to analyse cellular effects of PARP inhibition.

Overall, I am impressed by the quality of the data obtained by QIBC analysis. The sensitivity indeed is better when compared for instance to flow cytometry analysis. I am also sure that using this technique biological mechanisms can be uncovered that cannot be solved using less advanced techniques.

However, to be honest, in this manuscript, I have not seen cellular phenotypes upon PARP inhibitor treatment, that were not observed previously, or could not be done using

more conventional analyses. We already knew that PARP inhibition requires its trapping activity to induce cytotoxicity (Murai, Pommier), and that PARP inhibition requires S-phase to induce DNA lesions. Also the recruitment of Rad51 (dependent on BRCA1/2) is in line with previous findings, and the sensitizing effects of ATR inhibition to PARP inhibition have been described previously. The aspects that are relatively new are the role of PARG and the effects of individual PARP isoforms, although in depth mechanistic insight for these observations is not provided. Further, the manuscript lacks technical details and proper controls, as indicated below.

Overall, to my opinion the current manuscript lack sufficient novelty and impact to warrant publication.

We were glad to read that this reviewer was impressed by the sensitivity of the approach and the quality of our data and acknowledged the advance over currently used techniques. As a technology manuscript, our main goal was indeed to critically assess and validate the approach against existing alternatives, building on the extensive knowledge on PARPi functions, and to provide new informative tools to study PARP biology and PARPi toxicity.

While we agree that our results are consistent with previous findings and models on PARPi toxicity, we would like to point out several key advantages, which we may have missed to explain well enough in our original manuscript, and which we believe go beyond the current state-of-the-art. For instance, to our knowledge previous assays have not assessed PARP trapping and DNA damage signaling simultaneously in the same cell population and in a cell cycle resolved manner (which we do based on co-staining of chromatin-retained PARP, γ H2AX and DAPI). Thus, it is not only more sensitive (please compare Extended Data Figure 4a and the new Extended Data Figure 4b), but has the clear advantage that DNA damage signaling in different cell cycle phases can directly be correlated with PARP trapping. Showing PARP trapping and DNA damage signaling in the same cell population and without the need to synchronize and release cells (which comes with significant damage induction itself, please see below) can therefore add direct evidence for and strengthen previously proposed models about PARP trapping-related damage during DNA replication (please see also point 8 by reviewer 3 concerning this theme). Along these lines, we have extended our analyses of simultaneously measuring PARP1 trapping and DNA damage signaling to H₂O₂ as a second well-known activator of PARP1 (Extended Data Figure 5a and 5b), and performed a proof-of-principle combinatorial drug screen in which we measure PARP1 trapping together with γ H2AX and EdU in multiwell format (Extended Data Figure 5c and 5d). Similarly, RAD51 foci were quantified at the single cell level and can be directly correlated to γ H2AX levels and cell cycle position. These read-outs therefore have the potential to aid basic research on PARPi functions, but also to improve the design of multi-dimensional screens. We have indicated more clearly in the revised manuscript text whenever multiple cellular responses were assessed simultaneously in the same cell population.

In addition to presenting and critically evaluating the technique and compare it to other currently used methods, we also extended several aspects of our work to reveal how this technique can be used to generate new biological insights:

- 1) We extended the analysis of the contribution of different PARP family members (PARP1, PARP2, PARP3) to different PARPi (veliparib, olaparib, talazoparib). These in vivo measurements revealed that PARP1 and PARP2 contribute to veliparib- and olaparib-induced DNA damage signaling, but that talazoparib-induced DNA damage signaling is alleviated mainly by depletion of PARP1

(Extended Data Figures 7a and 7b). These results, obtained in an isogenic system, extend previous work (e.g. Mol Cancer Ther. 2014 Feb;13(2):433-43.), and the presented short-term microscopy-based analyses provide an alternative to existing survival and biochemical fractionation experiments.

- 2) We also extended the analysis of the partial rescue we observed when depleting PARG. While we found that PARG knockdown did not alleviate chromatin retention / trapping of either PARP1 or PARP2, it did rescue PAR levels in the presence of olaparib. This was accompanied by reduced formation of RAD51 and also BRCA1 foci in S-phase cells, suggesting that fewer lesions were formed in these cells. While elucidating the exact nature of these lesions is beyond the scope of this manuscript, our results suggest that PARP inhibition, even at a 10 μ M concentration, is incomplete in cells, and that PAR formation can be rescued and PARPi toxicity alleviated by down-regulation of PARG. The new data are presented in Extended Data Figures 8 and 9.
- 3) Based also on the recommendation by reviewer 4 to assess the compatibility of the presented technique for high-content screening, we performed proof-of-concept siRNA and drug screens in multiwell format. This not only extended the applicability of our approach, but it also revealed that loss of SKP1 and CUL1, two components of the SCF ubiquitin ligase complex, specifically sensitized cells to PARPi. Based on this result we tested whether pevonedistat, a potent NEDD8/SCF inhibitor in clinical trials for cancer treatment, would synergize with PARPi. Since this was indeed the case, we assessed PARP1 trapping and found elevated PARP1 on chromatin in the presence of pevonedistat. Consistently, loss of PARP1 rescued the pevonedistat-induced DNA damage and the associated cell cycle arrest, strongly indicating that the presence of PARP1 is required for this newly discovered drug interaction. While characterization of the exact mechanism underlying this interaction is a project on its own, these new additions demonstrate how multivariate read-outs upon short-term experiments can predict genetic interactions and drug synergism and how they can be upscaled for screening purposes, and the results open the door for future work on the interplay between SCF ubiquitin ligases, PARylation and PARP1 trapping. This new set of data is presented in Figure 7 and Extended Data Figure 14.

Specific Points:

In the abstract, a few claims are made that are not substantiated by the results: 'our approach can be adapted to predict outcomes' and '...ultimately stratify cancer patients'. If anywhere, these statements should be in the discussion.

We agree and have revised the abstract accordingly.

If the authors want to make a statement that QIBC is required to predict outcome to a range of parp inhibitors at early time-points, then a comparison to an alternative technique is required. Previously, the various PARP inhibitors were already compared by the Pommier lab, in assays that only required few hours of PARP inhibitor treatment, which also predicted cytotoxicity of the compounds.

As outlined also below, we included additional controls to validate short-term QIBC measurements by other techniques (e.g. synchronization-release experiments, biochemical fractionation). To directly assess the effects of different PARPi on cell proliferation and viability in our conditions, we employed high-content live imaging to

follow populations of GFP-H2B U-2 OS cells over time in the absence or presence of PARPi. Image-based software-assisted quantification of cell counts at 24 h, 48 h, and 72 h recapitulated the short-term QIBC results, with olaparib being more potent than veliparib and veliparib being more potent than PJ-34 (Extended Data Figure 1h). We conclude that the differences in DNA damage signaling with different PARPi (Figure 1b) translate into corresponding differences in cell viability.

Indeed, PARPi toxicity was previously assessed in short-term assays such as biochemical fractionation to check PARP trapping. However, to our knowledge the previously reported assays are less sensitive (e.g. we observe 5-10 fold increases in PARP1 trapping, Extended Data Figures 4a and 4b), do not provide single cell or cell cycle information, and are more difficult to upscale for screening purposes. In our revised manuscript we show a proof-of-concept combinatorial drug screen in multiwell format, in which chromatin-bound PARP1, DNA damage signaling and EdU incorporation are assessed simultaneously (Extended Data Figures 5c and 5d). This revealed interesting similarities but also differences between drug combinations (e.g. MMS vs. H₂O₂) and demonstrates how the presented approach can be used for informative phenotypic single and combined drug screens. For these reasons we believe that multi-dimensional analyses by QIBC provide additional benefits, which can be exploited to complement existing assays to assess PARPi responses and predict cellular outcomes.

The hypercondensation that is observed upon combined treatment with olaparib and ATRi may be due to prolonged time in mitosis. To control for this, the authors include MG-132 treatment. However, mitotic hypercondensation is time-dependent, and so the time of treatment should be taken into account. In this case, the MG-132 treatment was done much shorter than the other treatments. For a fair comparison, the time that cells spend in mitosis should be taken into consideration. Related to the point above, the NCAPH2 localization to mitotic chromosomes needs to be performed in all treatments, to see if this is lost specifically in the combined treatments. Also, these results should be quantified from multiple cells, and statistics should be applied. Regardless, this might be an epiphenomenon.

We agree with this reviewer that alternative explanations cannot be ruled out and that for the differential localization of NCAPH2 it is difficult to unambiguously discriminate cause from consequence. Since also reviewer 3 considered these data irrelevant for the manuscript, we removed the NCAPH2 and MG-132 results from the revised manuscript. We would prefer to keep the QIBC data on mitotic chromosome area and DNA condensation, however, simply to show how such measurements of morphology and DAPI intensity can be used to reveal phenotypes in rare sub-populations (such as mitotic cells), which would be easily lost when analyzing cell population averages. We extended these experiments by H3pS10 co-staining, as suggested by reviewer 2, to show that our analyses are indeed focused on mitotic cells. In the revised manuscript we discuss these data more carefully and we agree that the robust increase in condensation may be due, at least partially, to the prolonged time in mitosis. We would like to thank the reviewer for pointing this out.

The observation that combined PARPi and ATRi treatment leads to increased cytotoxicity is stated to 'likely' be due to mitotic aberrancies. Firstly, it is not clear in which cell type these experiments were performed. If performed in a HR-proficient cell line, ATR inhibition may induce HR deficiency which could underlie the observed effects. Also, the current experiments do not point at which of the many functions of ATR

underlie the observed effects. Loss of cell cycle checkpoint behaviour? DNA repair? Fork stability?

Also here we agree that ATR has multiple functions during S-phase progression (and even beyond) and that these may collectively contribute to the increased cytotoxicity in the presence of PARPi. We performed these experiments in HR-proficient U-2 OS cells and, as we show in Extended Data Figure 13b, ATRi indeed leads to a defect in PARPi-induced RAD51 foci formation, suggesting that ATRi and PARPi synergize in S-phase. However, our data further show that ATRi treatment increases the percentage of PARPi-exposed cells that progress into mitosis, where they experience high levels of DNA damage (Extended Figures 11a-d). Thus, the clinical potential of ATRi is likely coupled to the multiple important functions of ATR during S-phase progression and checkpoint activation. While specific assays such as DNA fiber experiments to measure fork speed and stability, or separation-of-function mutations can be used to study specific ATR functions, our approach looks at the functional consequences of ATR inhibition for proliferating cell populations more globally, and thus complements existing techniques at a level that integrates cell cycle dynamics and checkpoint activation with DNA damage signaling. With this in mind, we extended our analyses further and now show that ATRi/PARPi also leads to greatly elevated levels of 53BP1 nuclear bodies in G1 cells marking inherited DNA lesions (Extended Figure 11e). Thus, not all cells experience catastrophic damage immediately in mitosis, but those, which exit mitosis, do so with high levels of DNA damage originating from the previous S-phase. We show these data to illustrate how QIBC can capture the dynamics of DNA damage formation across the cell cycle in a sensitive and quantitative manner, and we would like to thank this reviewer for his/her comments, which encouraged us to look at the ATRi/PARPi interaction in a more comprehensive manner.

-knockdown should be verified when used in experiments (Fig 2/5). WB or qPCR data should be included.

We performed the requested qPCR experiments to confirm the knockdowns and included also the new PARP1/2/3 triple depletion (Extended Data Figures 2e and 7c).

-it is unclear from the figure legends or figures which cells are used, and which concentrations of drugs were applied.

We now more clearly indicate cell lines and compound concentrations in the figures, the figure legends, and in the methods section.

-Figure 6: mitotic duration should be quantified. Please quantify G2 arrest. Please indicate G2 arrest separately from S/G2 arrest.

We extended these 2-dimensional cell cycle profiles (based on DAPI and Cyclin A) to 4-dimensional ones (based on DAPI, Cyclin A, EdU and H3pS10) to compare olaparib to talazoparib. In contrast to 1-dimensional cell cycle profiles based on DNA content only (e.g. by PI/Hoechst/DAPI), which leave some uncertainty when trying to discriminate G1 from early S, G2 from late S, and M from G2, these 4-D profiles allowed us to properly gate for G1, S, G2 and M. Consistent with our 2-D data, olaparib reduced replication speed and this was associated with an accumulation of cells in S and G2 and with a decrease in the mitotic population. Talazoparib had a more severe effect, with replication being strongly compromised, resulting in an accumulation of cells directly in

S-phase. Also here, this was accompanied by a strong decrease in the mitotic cell population. These new data are provided in Extended Data Figure 10.

Minor points:

-The U2-OS cell lines does not strike me as a very logical cell line to uncover PARPi effects, since the authors start the paper to identify how PARPi actually cause cytotoxicity, because this may guide treatment decisions and patient selection. A more logical choice would be ovarian/breast cancer models with/without HR defects.

We use U-2 OS cells as model cell line, mainly because they have negligible DNA damage signaling in unchallenged conditions and show normal replication and cell cycle profiles without endoreduplication (see our untreated conditions). This makes these cells well suited for sensitive, image-based measurements of changes in DNA damage signaling and cell cycle progression. Based on this reviewer's comment we validated our approach in a BRCA-mutated, HR-defective and PARPi-sensitive breast cancer cell line (MDA-MB436) compared to a PARPi-resistant breast cancer model (HCC1143). Reassuringly, we could detect PARPi-induced DNA damage signaling at early time-points in MDA-MB436, but not in HCC1143 cells. We included these data as Figure 2d.

-Lynparza is a brand name and is spelled with a capital, all the PARP inhibitor names (olaparib, talazoparib etc) are spelled with small fonts.

We thank the reviewer for pointing this out and changed the capitals in PARP inhibitor names to lower font as requested.

-Page 5: 'and quantified all relevant parameters of PARPi toxicity'. This is a strong overstatement. There are multiple parameters of PARPi toxicity that may be important for PARPi toxicity (eg P-glycoprotein expression as the authors earlier indicate).

We agree that this was an overstatement and changed the text accordingly.

-In the introduction, it is mentioned that Lynparza is approved, and other PARP inhibitors are in late clinical testing. Actually, Rubraca and Zejula have also been approved.

We thank the reviewer for this correction and modified the text accordingly.

Reviewer #2 (Remarks to the Author):

Here J. Michelena et al. develop a sensitive single cell assay to assess cell cycle specific responses to PARPi after relatively short term treatments. This is complementary (and perhaps more sensitive and quantitative) than current protocols which measure population averages in long-term clonogenic survival experiments. A few nice examples of the utility of the method are given, including how ATRi synergizes with PARPi, measurements of PARPi induced PARP1 trapping, as well as the demonstration that PARG-mediated turnover of PAR contributes to PARPi toxicity. This is a solid report which will be useful for the scientific community.

We would like to thank also this reviewer for the constructive comments on our work and were glad to read that he/she seems to appreciate the advances associated with the presented single cell approach, the sensitivity and quantitation possibilities, and the benefits associated with short-term responses measured in high-throughput, and that publication of our report will make a useful contribution for the scientific community.

They should address the following questions prior to publication:

1) They demonstrate that defects in HR sensitize to PARPi. They note differences in gamma H2AX at 16 and 48 hours post-Parpi treatment (Figures 2a-c). Why don't they see differences earlier- after all, in Figure 1 they see a PARPi responses in S phase already by 15 minutes. If they don't see a difference in DDR signaling between WT and HR deficient early on, they should explicitly show and explain this.

Indeed, when performing BRCA1 and BRCA2 knockdown, we do not observe obvious differences in S-phase γ H2AX at earlier time-points. We can only speculate that this may be related to incomplete knockdown by siRNA with residual BRCA function being sufficient to protect cells transiently from PARPi toxicity as measured by γ H2AX induction. As this would be a rather technical reason, we feel that it would not add too much to the manuscript to show data at earlier time-points, but we do mention it explicitly in the revised manuscript (page 10). Importantly, however, we now show data of BRCA1-mutated breast cancer cells, in which we detect PARPi-induced γ H2AX at 4 h and 8 h, in comparison to resistant cells, which are protected from DNA damage (Figure 2d). Although these cells are non-isogenic, the new experiments demonstrate that PARPi-induced DNA damage signaling in BRCA1-deficient cells can be detected and quantified by automated high-content microscopy at early time-points.

2) PARPi and formaldehyde toxicity synergize- Figure extended Data Fig. 3b. Is this only in BRCA2 deficient cells, or also WT or BRCA1 deficient?

We apologize for not having explained this well enough in our original manuscript. These experiments were performed in BRCA1/2-proficient U-2 OS cells and were based on previous findings showing that formaldehyde exposure (using identical conditions as in our manuscript) leads to destabilization of BRCA2 and, to a lesser extent, also of BRCA1 and RAD51 (Cell. 2017 Jun 1;169(6):1105-1118.e15.). This results in an *induced* haploinsufficiency, and we aimed to test whether this situation would synergize with PARPi. To control for the effect of formaldehyde in U-2 OS cells we assessed RAD51 foci and observed a partial defect as would be expected based on the published study. We provide these data as Figure 1 to the reviewers and have clarified the formaldehyde-

induced HR defect in the manuscript text (page 11). If this reviewer feels that these additional data should be shown in our manuscript, we will be happy to include them.

Figure 1 for the reviewers: Formaldehyde exposure reduces PARPi-induced RAD51 foci formation. U-2 OS cells were treated as indicated with 0.1 mM formaldehyde, 10 μ M olaparib, or a combination of both for 4 hours and RAD51 foci formation was assessed by QIBC. The results are consistent with compromised HR function upon formaldehyde exposure as published previously (Cell. 2017 Jun 1;169(6):1105-1118.e15.).

3) How does depletion of PARG (which increases PAR levels) and depletion of PARP1/PARP2 (which reduces PAR levels) both result in reduced levels of DDR signaling? This is not intuitively obvious. Does reduction of PARG reduce PARPi induced trapping? Does PARG depletion also result in cisplatin resistance?

Based on these suggestions we investigated PARP trapping in PARG-depleted cells in our system. Interestingly, neither PARP1 nor PARP2 showed decreased trapping in PARG-depleted cells (Extended Data Figures 8b and 8c). We therefore considered the hypothesis that loss of PARG would influence replication fork reversal, a process in S-phase cells, which is PAR- and PARP1-dependent and which protects cells from replication damage (Nat Struct Mol Biol. 2012 Mar 4;19(4):417-23.). In line with this published work, we observed that PARPi rescued replication fork speed in the presence of low doses of camptothecin and lead to unrestrained fork progression. PARG depletion did not reverse this phenotype, however, making it unlikely that the rescue of DNA damage signaling by PARG knockdown is due to restored replication fork reversal (Figure 2 for the reviewers). Nevertheless, we observed that PARG knockdown rescued PAR levels in olaparib-treated cells (Extended Data Figure 8d). The restored PARylation was associated with alleviated formation of RAD51 and BRCA1 foci upon PARPi (Extended Data Figures 9a and 9b). While the exact mechanism of siPARG-relieved PARPi toxicity needs further studies, these results suggest that PARP inhibition, even when using 10 μ M olaparib, is not complete in cells and that PAR degradation by PARG contributes to reduced PAR formation and PARPi toxicity. Consistently, upon PARG knockdown we observed markedly increased levels of chromatin-bound XRCC1, one of the many PAR-binding protein that is recruited to sites of DNA damage in a PAR-dependent manner (Figure 3a and 3b for the reviewers). Co-depletion of XRCC1 only very mildly restored PARPi sensitivity in PARG-depleted cells (data not shown) and we therefore believe that additional PAR-dependent functions and protein recruitments are restored in olaparib-treated PARG-depleted cells and may function collectively to alleviate PARPi toxicity.

Figure 2 for the reviewers: U-2 OS cells were transfected with siRNA against PARG and exposed to olaparib as indicated and DNA fiber experiments were performed with camptothecin (CPT) being added together with the second label. The IdU/CldU ratio indicates replication fork slowdown in the presence of CPT. This is rescued by olaparib, irrespective of PARG depletion. 100 fibers per condition were scored.

Figure 3 for the reviewers: (a) U-2 OS cells were transfected with siRNA against PARG as indicated, pre-extracted on ice in 0.2% Triton X-100 for 2 min to remove soluble, non-chromatin-bound proteins, and stained for XRCC1 and DNA content. (b) U-2 OS cells were transfected with siRNA against PARG as indicated, treated or not for 1 h with 0.1 mM H₂O₂, pre-extracted on ice in 0.2% Triton X-100 for 2 min to remove soluble, non-chromatin-bound proteins, and stained for XRCC1 and DNA content.

We also assessed the effect of PARG depletion on cisplatin-induced DNA damage signaling. However, we did not observe a rescue upon PARG depletion (Figure 4 for the reviewers). Although these results provide additional specificity to the observed rescue of PARPi toxicity by PARG knockdown, we felt that they may go beyond the main focus of our current manuscript and we therefore provide them for the reviewers only. If this reviewer feels that these data should be shown in our manuscript, however, we will be happy to include them.

Figure 4 for the reviewers: U-2 OS cells were transfected with siRNA against PARG as indicated, treated with cisplatin (5 μ M), olaparib (10 μ M) or a combination of both for 8 hours and stained for γ H2AX and DNA content. Cell cycle resolved γ H2AX profiles are shown.

4) Combined PARPi/ATRi lead to gamma H2AX specifically in mitotic cells. This was examined visually by DAPI staining (Figure 7). They should also quantify by QIBC using a mitotic marker (H3S10p and MPM2-p).

We performed the requested experiments and show in the new Extended Data Figure 11 that cells with high mean DAPI intensity are highly positive for the mitotic marker H3pS10, and that H3pS10-positive mitotic cells accumulate high levels of γ H2AX upon combined treatment. Although we do not show cell images for each of the QIBC experiments, we would like to point out that all quantifications are, as a default routine during the image acquisition and analysis, coupled to a visual inspection of the relevant cellular phenotypes (i.e. the numerical data shown in our scatter plots are linked to the original images and allow for visual validation of all cellular features).

Reviewer #3 (Remarks to the Author):

PARP inhibitors (PARPi) have been utilized in the clinical treatment for Stage IV ovarian cancers with BRCA mutations. In this study, the authors used fluorescence microscopy and quantitative image-based cytometry to examine the defects of DNA damage response induced by PARPi treatment. Although this novel approach may be useful for analyzing the efficacy of PARPi, the study lacks mechanistic insight. Furthermore, the overall quality of the study does not reach the level of Nature Communications.

We were glad to read that this reviewer appreciates the novelty of our approach and its potential to study PARPi effects. For a technology-focused manuscript, our aim was primarily to evaluate the benefits of multivariate quantitative high-content imaging compared to other techniques currently used to assess PARPi toxicity. Clearly, for deep mechanistic insights into the types of DNA lesions induced by PARPi and how the cellular repair machinery deals with them, our technique would have to be combined with dedicated orthogonal methods (e.g. electron microscopy-based studies of replication intermediates, single molecule assays to assess replication fork speed and stability, proteomics to identify PARylation targets, biochemical assays to validate PAR acceptor sites, etc.). Conversely, biochemical *in vitro* studies often benefit from an *in vivo* counterpart, which assesses the cellular responses and functional consequences. It is this latter aspect, for which we believe that the presented technique can be a highly informative tool, which is sensitive, quantitative, has single cell and sub-cellular resolution, and which can be conducted in high throughput for screens, as we demonstrate in our revised manuscript. We hope that our revised manuscript with the extended biological insights, the quality of the data, and the additionally included control experiments will convince this reviewer of the utility of the technique and the usefulness for the scientific community.

Specific comments:

1. In the study, the authors show that PARPi treatment causes DNA damage in the S phase. However, it is unclear how PARPi induces DNA damage exclusively in the S phase, and what type of lesions is caused by PARPi treatment.

We agree with this reviewer that the type of lesions remains obscure. Despite more than 10 years of intense research on PARPi toxicity and synthetic lethality following the two Nature publications by Bryant et al. and by Farmer et al. it is not clear how the DNA structures look that give rise to PARPi toxicity. In fact, it is not unlikely that multiple different structures collectively underlie PARPi toxicity (reversed replication forks, which may provide an entry point for enzymatic degradation, PARP-DNA adducts, which may have to be removed for the replication fork to pass by, unrepaired single-strand breaks, which can be converted to double-strand breaks in the context of replication, and potentially others). In different contexts, these may have different relative importance. Our ambition was to present and evaluate novel tools, which can help assess the cellular consequences of impaired PARP functions (including DNA damage signaling, cell cycle arrest, PARP trapping, etc.). We believe that such tools will be of great benefit for the scientific community and will aid the validation/rejection of existing models, allow for phenotypic discovery screens and eventually help patient selection based on multivariate *ex vivo* analyses of cancer cells.

2. Fig. 1b lacks a mock treatment control. A large field of images with gH2AX positive cells should be shown.

We had controlled that DMSO treatment, the proper mock control for our PARPi experiments, does not lead to detectable DNA damage signaling or any signs of impaired cellular fidelity. We provide these data for the reviewers only (Figure 5 for the reviewers), and would be happy to add them to the manuscript according to this reviewer's recommendation. As requested, we provide a larger field of images to accompany our analyses in Figure 1.

Figure 5 for the reviewers: Asynchronously growing populations of adherent U-2 OS cells were left untreated, treated with DMSO or 10 μ M olaparib in DMSO (both 1:1000) for 4 h, fixed and stained for DNA content (DAPI) and the genotoxic stress marker γ H2AX. Scatter plots depict mean γ H2AX and total DAPI intensities per nucleus.

3. In Fig. S1d, the authors did not find obvious gH2AX using Western blot in unsynchronized cells. However, the authors should harvest the S phase cells and analyze gH2AX in the S phase population.

One advantage of the presented technique is that it allows evaluation of cell cycle phase specific responses without having to synchronize cells. As requested by the reviewer we validated our results by synchronizing cells in G1 and releasing them into S-phase. From these cells we extracted histones to probe for γ H2AX by Western blot. Indeed, under such conditions we observed an increase in γ H2AX, consistent with our QIBC analysis (Extended Data Figure 1e,f). In the same experiment, we assessed γ H2AX by QIBC in thymidine-arrested cells and we provide the results below for the reviewers. Of note, the thymidine arrest induced significant amounts of DNA damage (Figure 6 for the reviewers) and this may obviously interfere with downstream analyses of damage signaling and cytotoxicity. We therefore believe that analyzing cellular drug responses in a cell cycle resolved manner without the need to synchronize, as presented in our manuscript, has clear benefits over the current biochemical alternatives.

Figure 6 for the reviewers: U-2 OS cells were synchronized with thymidine for 20 hours and stained for DNA content (DAPI) and the genotoxic stress marker γ H2AX. Scatter plots depict mean γ H2AX intensities as a function of the cell cycle. For the same cell populations one-dimensional cell cycle profiles based on DAPI staining are shown below. Untreated and olaparib (10 μ M, 6 hours) treated asynchronously growing cells are shown as a comparison.

4. Statistical analysis is lacking in Fig. S1e. It is unclear if ola treatment has any significant effect.

As we are comparing typically several thousand cells per condition, the differences are significant in statistical tests even if the differences are seemingly mild and if conservative tests are used, e.g. not assuming a Gauss distribution (e.g. for 1a, S1c, S1g: $p < 0.0001$, Mann-Whitney). To assess the robustness of our approach we therefore focused on the least obvious difference in our dataset and tested the minimal number of cells required to reach significance in statistical tests. The relevant data are the ones in which we compare untreated cells to cells treated for just 15 minutes with olaparib and in which only a relatively small cell population is analyzed (Figure 1c). Across the whole cell population (i.e. not focusing the analysis on the S-phase population only), 400 cells were enough for a p-value of < 0.05 . We included these analyses in the revised manuscript in Figure 1c.

5. In Fig. 1C, the authors used 10 μ M of 5 different PARPis to examine γ H2AX. However, the IC50s of these PARP1 are generally below 1 μ M. And the IC50 is different in different cancer cell lines. Thus, the conclusions are invalid.

We respectfully disagree that our conclusions are invalid. We agree, however, that they cannot be generalized per se to other cell lines or other biological systems. Our aim was to compare different PARPi in the same cell line at the same concentration in a short-term assay. We now validate the differences we observed in DNA damage signaling by

cell proliferation assays for PJ-34, veliparib and olaparib, and these proliferation experiments are aligned with our short-term γ H2AX measurements (Extended Data Figure 1h). The concentration of 10 μ M has been used in countless publications, and the IC50 of olaparib was shown to vary from low nM to 21.7 μ M (!) in different cell lines (Oncotarget. 2017 Jun 20;8(25):40152-40168.). Nevertheless, we repeated the analysis with a 10-fold lower concentration and, although the response is expectedly less pronounced, the results nicely confirmed the observed trend. We provide the results as Figure 7 for the reviewers and hope that this reviewer agrees that such short-term assays can provide a viable, screening-compatible alternative to and complement long-term proliferation and survival assays.

Figure 7 for the reviewers: U-2 OS cells were treated with the indicated PARPi for 4 h and stained for γ H2AX and DNA content. γ H2AX levels are shown as a function of cell cycle progression.

6. In Fig. S2, the authors treated cells with ola for 8 hours and observed obvious RAD51 foci in the S phase cells. It indicates that numerous DSBs have been generated. However, it is unclear how cells were still evenly distributed in different cell cycle phases and did not activate cell cycle checkpoints.

We completely agree that it is informative to assess cell cycle effects under these conditions. While 8 hours is a relatively short time to expect strong changes in cell cycle distribution (e.g. measured by conventional 1-dimensional DNA content-based cell cycle analysis by flow cytometry), we indeed measured, using EdU labeling for 2-dimensional cell cycle profiling, that replication speed was reduced and cells accumulated in S-phase (Extended Data Figure 2c). Thus, and as predicted by the reviewer, RAD51 foci formation and γ H2AX signaling in S-phase cells, which we measured in the same cells used for the EdU/DAPI analysis, is associated with a slow-down of S-phase progression.

7. In Fig. 2B, in the BRCA1 or BRCA1 knockdown cells, PARPi treatment induced cell accumulation in the G1 and G2 phases. What is the underlying mechanism? In Fig. 2C, When FANCD2 was down-regulated, cells were only arrested in G2/M with PARPi treatment. Thus, the underlying mechanism could be totally different from those in the BRCA1 or BRCA1 knockdown cells.

Here, our aim was not to dissect the mechanism of BRCA1/BRCA2 or FANCD2-related PARPi hyper-toxicity, but rather to show that our approach can recapitulate known synthetic lethal relationships. Both, loss of BRCA1 and BRCA2 are known to be synthetic

lethal with PARPi (Nature 434, 913-917 (2005), Nature 434, 917-921 (2005)), and the same is true for loss of FANCD2 (Cancer Res 66, 8109-8115 (2006), Clin Cancer Res. 15;21(8):1962-72(2015)). We agree that the underlying mechanisms could differ, and detailed time- and cell cycle resolved experiments as the ones provided in our manuscript may in fact help to address such questions in the future.

8. In Fig. 3a. the chromatin-bound PARP1 should be validated with Western blot. In addition, the chromatin-trapped PARP1 is not correlated with gH2AX because gH2AX is only elevated in the S phase cells.

As requested by the reviewer we validated the trapping of PARP1 by biochemical chromatin fractionation. Although less sensitive, the results confirmed our QIBC measurements (Extended Data Figure 4b).

Indeed, the chromatin-trapped PARP1 is not correlated with γ H2AX, as the former occurs in all cell cycle phases whereas the latter preferentially forms in S-phase. An important advantage of our technique compared to the available alternatives is that we can look at these two parameters, PARP1 trapping and γ H2AX formation, in the same cells (based on co-staining) and in a cell cycle resolved manner. Thus, we can unambiguously conclude that, although PARP1 trapping is cell cycle independent, problems occur mostly in S-phase when DNA replication occurs. This is consistent with previous reports, as reviewer 1 pointed out, but to our knowledge it is the first time that within the same cell population PARP1 trapping, γ H2AX formation, and cell cycle position are assessed simultaneously and with high sensitivity and in a screening-compatible manner (see new Extended Data Figures 5c and 5d).

9. Fig. 4c is more problematic. The second sample should be same as the third sample in Fig. 3b. However, there are totally two different pattern of gH2AX.

We are grateful to this reviewer for noticing this mistake in the figure assembly. One experiment had been done with 1 h treatment, the other with 4 h treatment. While we observe PARP1 trapping more evidently upon 4 h treatment, the development of the phenotype – PARPi-induced PARP1 trapping in a cell cycle independent manner and γ H2AX signaling in S-phase – is better visualized after 1 h. In the new Figures 3d and 3e we now show PARP1 profiles (chromatin-bound) together with corresponding γ H2AX profiles from the same cells after 1 h treatment to better illustrate this point. Figure 4c has been corrected and the difference in the treatment duration is explained in the main text and stated in the corresponding figure legends.

10. In Fig. 5b, with ola treatment, PARP1/2 were still trapped in DNA lesions. How can knowdown of PARG suppress gH2AX. What is the underlying mechanism?

Also reviewer 2 was intrigued by our finding that loss of PARG partially rescues PARPi toxicity. Based on these comments we investigated PARP trapping in PARG-depleted cells in our system. Interestingly, neither PARP1 nor PARP2 showed decreased trapping in PARG-depleted cells (Extended Data Figures 8b and 8c). We therefore considered the hypothesis that loss of PARG would influence replication fork reversal, a process in S-phase cells, which is PAR- and PARP1-dependent and which protects cells from replication damage (Nat Struct Mol Biol. 2012 Mar 4;19(4):417-23.). In line with this published work, we observed that PARPi rescued replication fork speed in the presence of low doses of camptothecin and lead to unrestrained fork progression. PARG depletion

did not reverse this phenotype, however, making it unlikely that the rescue of DNA damage signaling by PARG knockdown is due to restored replication fork reversal (Figure 2 for the reviewers, please see above). Nevertheless, we observed that PARG knockdown rescued PAR levels in olaparib-treated cells (Extended Data Figure 8d). The restored PARylation was associated with alleviated formation of RAD51 and BRCA1 foci upon PARPi (Extended Data Figures 9a and 9b). While the exact mechanism of siPARG-relieved PARPi toxicity needs further studies, these results suggest that PARP inhibition, even when using 10 μ M olaparib, is not complete in cells and that PAR degradation by PARG contributes to reduced PAR formation and PARPi toxicity. Consistently, upon PARG knockdown we observed markedly increased levels of chromatin-bound XRCC1, one of the many PAR-binding protein that is recruited to sites of DNA damage in a PAR-dependent manner (Figure 3a and 3b for the reviewers, please see above). Co-depletion of XRCC1 only very mildly restored PARPi sensitivity in PARG-depleted cells (data not shown) and we therefore believe that additional PAR-dependent functions and protein recruitments are restored in olaparib-treated PARG-depleted cells and may function collectively to alleviate PARPi toxicity.

11. Ola does not suppress or trap PARP3. How can PARP3 knockdown rescue the γ H2AX phenotype induced by ola treatment in Fig. S5c.

The effect of PARP3 knockdown on γ H2AX was very mild. We revisited this issue using a larger panel of PARPi and doing single and combined knockdowns of PARP1, PARP2 and PARP3 and the results overall agree with this reviewer's point and indicate a contribution of PARP1 and PARP2 but not PARP3. These data are provided as new Extended Figure 7b.

12. Data in Fig.7 are irrelevant with other data in the rest of the manuscript.

Also reviewer 1 brought up justified concerns regarding the development of the condensation phenotype, and we therefore removed the NCAPH2 data from the revised manuscript. The synergy between ATRi and PARPi, which is explored also in the clinics, inspired us to design a targeted siRNA library with ATR/CHK1 activators, checkpoint proteins, and cell cycle regulators and perform a high-content imaging-based screen to identify, using a multi-dimensional read-out with single cell resolution, novel functional interactions with PARPi. We thus extended our manuscript towards this angle, which we feel is indeed better connected to the rest of our manuscript and identifies a novel, potentially very relevant connection between PARPi and the NEDD8/SCF inhibitor pevonedistat.

Reviewer #4 (Remarks to the Author):

The authors present a well written manuscript with an evident logic progression. The presented method, Quantitative Image Based Cytometry (QIBC), allows for the analysis of PARPi toxicity, on sub-population resolution, by the use of microscopy based single-cell analyses combined with cell cycle staging of an unsynchronized cell population. The applicability of QIBC for the comparison of individual PARP inhibitors and combination treatments in a cell cycle resolved manner is extensively demonstrated.

Furthermore, the authors elegantly exploit the possibilities of sub-cellular analysis, provided by microscopy based methods in contrast to flow cytometry, by extending their analysis from nuclear average values to the detection of nuclear RAD51 foci.

The presented method has been validated against state of the art methods by including the analysis of PARP inhibitor toxicity, PARP enzyme trapping and DNA damage response. A clear advantage in sensitivity and required treatment duration has been demonstrated and is attributed to the technological and methodological improvement achieved by single cell analysis, cell cycle staging and high throughput microscopy.

We were glad to read that this reviewer considers our manuscript well written and logically developed and appreciates the clear advantage in sensitivity and required treatment duration as well as the informative value of high content imaging with single cell and sub-cellular resolution. This is exactly what we were hoping to convey to the scientific community.

The manuscript is, however, a bit of a chimera, one part is the presentation of a new method and the other part is trying to elucidate PARP biology in cell cycle, disease and therapy. Both parts are interesting, but none is totally convincing:

1) The new QIBC method is not as novel as claimed. At the end of the day, the authors describe the cellular response to PARP inhibition in combination with siRNA mediated manipulation of a test cell line. The efficiency and specificity of their siRNA treatments is not shown. The cellular response is determined mostly pairwise, which is OK, but not novel. Here real multi-parameter analysis would have been better and perhaps even revealing.

While QIBC has indeed been used in a few other settings before, it has not been applied to comprehensively analyze PARPi responses as performed here. Moreover, currently very few research groups worldwide perform microscopy-based cell cycle staging and software-assisted feature extraction in large populations of cells to assess cellular responses to genotoxic stress. The current state-of-the-art, irrespective of the quality of research and the journal in which it is published, is to assess (mostly manually) the percentage of cells with more than a certain arbitrary number of sub-nuclear foci, and in the majority of cases cell cycle position is not accounted for. Therefore, there is a great need for advanced techniques to facilitate quantitative, multi-dimensional image-based cytometry and for providing examples and technical descriptions in relevant biological contexts. We hope that our work can contribute to an aspired trend of more quantitative cell biology and provide the examples and protocols needed to employ automated high-content imaging as highly informative research tool, be it in individual assays or in high-throughput screens.

Knockdown efficiencies for BRCA1, BRCA2, FANCD2, PARP1, PARP2 and PARP3 were now confirmed by qPCR, as also suggested by reviewer 1 (new Extended Data Figures 2e and 7c). We apologize for not having indicated clear enough where multiple parameters

were evaluated in the same cell populations. Since multi-dimensional analyses are difficult to visualize on 2-dimensional paper, we sometimes split them into different data panels. For example, we assessed PARP trapping, cell cycle and γ H2AX signaling in the same cells (Figure 3). Similarly, we quantified RAD51 foci, γ H2AX levels, DNA content and EdU incorporation in the same cells (Extended Data Figure 2c), and we performed 4-dimensional cell cycle staging of cell populations based on DAPI/EdU/CyclinA/H3pS10 (Extended Data Figure 10). In all these experiments we additionally assessed and evaluated cell count, nuclear area, nuclear morphology, nuclear circumference and circularity factor, etc. In the newly added screen presented in Figure 7 we measured more than 2 million parameters in more than 200'000 single cells. While spectral overlap indeed puts a limit to how many markers can be combined in one experiment, we are currently working towards protocols to overcome this limitation with the goal to eventually be able to upscale also the number of markers that can be measured simultaneously.

2) In their abstract and introduction, the authors paint a rather dire picture of the state of the art and neglect far more advanced HTS/HCA studies. This being mostly a new methods manuscript, the authors should spend more energy describing prior work, e.g., when it comes to cell cycle staging the authors might want to mention similar protocols described before (e.g. the "Cell cycle staging of individual cells by fluorescence microscopy" Roukos et al., Nat Protoc. 2015 Feb;10(2):334-48). Currently, most of their citation are either PARP or their own.

We apologize for our negligence and agree with the reviewer that a more thorough discussion of previous work is appropriate. We rephrased the abstract and extended the introduction to better cover HTS/HCA analysis in other areas of biology, and refer to related protocols including the one by Roukos et al. in Nature Protocols.

3) The last sentence of the abstract: "Our approach can be adapted to predict outcomes of a variety of cytotoxic agents and ultimately stratify cancer patients." promises far more than the manuscript actually delivers and is confusing not to say misleading. This is the underlying but never substantiated claim throughout this manuscript. The test of other cytotoxic agents other than PARP inhibitors and patient derived cells with known lesions would have been necessary and helpful.

We removed this sentence from the abstract. We note, however, that we indeed test other cytotoxic agents (ionizing radiation, camptothecin, temozolomide, formaldehyde, ATRi), either alone or in combination, and we now also included patient derived cell lines with known lesions rendering them either PARPi sensitive or resistant (new Figure 2d).

4) Most measurements were performed with fixed and stained cells. Seeding "cells onto clean and sterilized glass coverslips", is rather low throughput and a bit outdated, certainly not suited for automated medium to high throughput screenings (HTS) and live cell readouts would have likely been more informative. Having not done so, certainly would have deserved some discussion.

One advantage of the presented approach is that it is suitable for both standard labs performing conventional IF experiments as well as for fully automated labs performing screens at a larger scale, and we hope that both can benefit from analyses similar to the ones described in our manuscript. We routinely perform these analyses in 96 and 384

well plates and, inspired by the reviewers' comments, we performed two proof-of-principle screens using single and combined drug treatments (Screen 1, Extended Data Figures 5c and 5d, measuring PARP1 trapping together with γ H2AX and EdU) as well as a targeted siRNA library with 3 independent siRNAs per gene, both in untreated and olaparib-treated conditions (Screen 2, Figure 7, measuring γ H2AX and EdU and a total of more than 2 million parameters in more than 200'000 cells). These new experiments not only demonstrate the applicability for high throughput RNAi and drug screens, but in addition shed new light onto PARP1 trapping in different conditions and revealed a novel functional interaction between the SCF machinery and PARPi, which we show is linked to PARP1 trapping and requires the presence of PARP1 to be effective (Figure 7 and Extended Data Figure 14). We would like to thank this reviewer for having inspired us to perform these experiments, which we consider as major advance of our revised manuscript.

We also performed high-content live cell imaging in multi well format to assess cell proliferation and survival upon PARPi. A condensed summary of the results is shown in Extended Data Figure 1h. While we agree that live cell readouts can provide additional information, such experiments critically rely on suitable markers. Unfortunately, this is currently not the case for γ H2AX, a PTM that requires antibodies for detection, and also for RAD51, which to our knowledge is very problematic to tag or overexpress. In such cases, where even endogenous protein tagging by CRISPR/Cas9 would disrupt protein function and would thus not work, dynamic information can be extracted by time-resolved experiments using fixed cells (e.g. as in Extended Data Figures 2a, 3c, 9a, 9b, Figure 5). We included a paragraph in the discussion, however, to point out the advantages of live cell imaging.

5) Having produced large data sets, the authors could and should have taken advantage of this resource and done some rigorous statistical testing/analyses. How robust is the method and when is a difference a significant difference?

As we are typically comparing several thousand cells per condition, the differences are significant in statistical tests even when conservative tests are used (not assuming a Gauss distribution) and when the differences are seemingly mild (e.g. for 1a, S1c, S1g: $p < 0.0001$, Mann-Whitney). To assess the robustness of our approach we therefore focused on the least obvious difference in our dataset and tested the minimal number of cells required to reach significance in statistical tests. The relevant data are the ones in which we compare untreated cells to cells treated for just 15 minutes with olaparib and in which only a relatively small cell population is analyzed (Figure 1c). Across the whole cell population, 400 cells were enough for a p-value of < 0.05 . We included these analyses in the revised manuscript in Figure 1c.

6) The coverage of prior publications on PARP biology is far from satisfying, as many insights into PARP function, interactions with cellular pathways and the effect of inhibitors are not cited/discussed. All conclusion about the role of PARPs, PARG, the cell cycle and homologous recombination are based on simple correlations and would either have to be discussed with much greater caution or studied much more thoroughly with systematic approaches and multiple complementing lines of evidence.

We indeed base several of our experiments on previous findings and the current knowledge on PARP biology and PARPi toxicity. We apologize for not having covered the prior publications comprehensively enough. In our revised manuscript we discuss

previous reports with more care and include references to prior works on PARP functions, PARP-related pathways, cell cycle and HR.

7) Did the authors evaluate the effects of PARPi treatment on non-responsive cell lines?

This was an excellent suggestion. We obtained two breast cancer cell lines, one BRCA1-mutated and HR-deficient, and one PARPi-resistant, and compared them side by side. The results, which show that DNA damage signaling is absent in the PARPi-resistant cells, are presented as new Figure 2d.

8) Is the method transferable to multiwell format? This would greatly improve throughput and screening capability. Especially in the context of predictive biomarker and drug screening on patient samples as well as the screening of large compound libraries in minimal volumes would be of great benefit.

Indeed, the method and all presented readouts are transferable to multiwell format. We thank this reviewer for raising this important point. We routinely perform such experiments in 96 and 384 well plates and among the most important additions to our manuscript are a targeted siRNA screen and a combinatorial drug screen, both performed in multiwell format, to identify new cytotoxic interactions with PARPi (Figure 7 and Extended Data Figures 5c and 5d). We hope that with these new additions this important point is now conveyed more clearly.

9) Could the authors please clarify whether images of individual confocal planes or z-projections over multiple planes were used for quantification?

We clarified this point in the methods section. The imaging is performed on a widefield screening microscope, which, despite the lack of confocal resolution, allows for faithful comparison of nuclear intensities and of sub-nuclear foci (e.g. the RAD51 foci shown in our figures). While there is a natural trade-off between image resolution and imaging speed / throughput, in our experience the robustness gained by an increase in the latter (allowing for cell cycle resolved measurements across thousands of individual cells per condition and of multiple conditions per experiment) often compensates for the slightly lower resolution associated with lower resolution objectives and non-confocal images. Obviously, this trade-off has to be considered and adjusted for new readouts.

10) Finally, the authors might want to revise sloppy jargon like e.g. at the end of the introduction: “by allowing replication-born lesions to prematurely enter mitosis” - lesions do not enter mitosis, but cells might.

We apologize for this imprecision and thank the reviewer for pointing it out to us. We have rephrased this sentence and tried to the best of our abilities to revise such jargon wherever we noticed it in our manuscript.

REVIEWERS' COMMENTS:

Reviewer #1 (Remarks to the Author):

In their revised manuscript, Michelena and co-workers have addressed my concerns.

Concerning the issue that although the methods are innovative, they have not identified unknown PARP biology: the authors now show that the method is more sensitive, and point to the fact that cell cycle data and PARP trapping can be obtained from the same cell population.

Also, additional work has been done to add biological insight. In this context, the role of PARP1, 2 and 3 have been compared, additional experiment of PARG depletion have been included, and a proof-of-concept siRNA screen has been included. To my opinion, the manuscript in its current form better underscores the potential use of the technique.

Also the phrasing in the abstract and body text better fits the results. In its present form, I think the manuscript is acceptable for publication.

Reviewer #2 (Remarks to the Author):

The authors have done a fine job and paper should be accepted

Reviewer #3 (Remarks to the Author):

The authors have adequately addressed my questions. The manuscript has been extensively revised and much improved. It would be better if the authors are able to discuss the DNA lesions induced by PARP inhibitors in S phase, which might be helpful for others using these inhibitors for basic research as well as clinical treatment for cancer patients.

Reviewer #5 (Remarks to the Author):

Michelena et al. have addressed all of the original points raised in the previous review in great detail and provided a substantial amount of new data that significantly improves the manuscript. In particular, the authors identify a novel synthetic lethal interaction between PARPi and pevonedistat in a new targeted high-content imaging screen using QIBC. However, there are still some outstanding issues that should be addressed before publication:

- As this is a technology manuscript the authors should provide much more detailed information in the Material and Methods section to allow other laboratories to follow their protocols. This is especially relevant when it comes to the QIBC analysis. E.g. while the color coding of the scatter dot blots results in easily understandable and beautifully presented Figure panels, it is not very accurate. For example in Figure 1b red dots are supposedly corresponding to cells with a γ H2AX "value" of 200. However, when looking at the blot, red dots can be found well below the 200 intensity on the y-axis (it is probably rather a range value). How exactly was the data plotted and the final graphs generated? How were the RAD51 foci counted? Furthermore, there is no information on the new targeted high-content imaging screen in the Material and Methods section and on how the 96 well and 384 well plate immunostaining experiments were performed.
- In the newly added target siRNA-based screen, the authors show that TIMELESS knock down leads to a dramatic increase in PARPi sensitivity. Besides being involved in the replication checkpoint, TIMELESS has been recently shown to interact with PARP1. The authors should briefly discuss this in their manuscript.

Michelena et al. *Analysis of PARP inhibitor toxicity by multidimensional fluorescence microscopy reveals mechanisms of sensitivity and resistance*

We would like to thank the reviewers for having re-evaluated our work and for supporting publication in Nature Communications.

Point-by-point response to the reviewers

Reviewer #1 (Remarks to the Author):

In their revised manuscript, Michelena and co-workers have addressed my concerns.

Concerning the issue that although the methods are innovative, they have not identified unknown PARP biology: the authors now show that the method is more sensitive, and point to the fact that cell cycle data and PARP trapping can be obtained from the same cell population.

Also, additional work has been done to add biological insight. In this context, the role of PARP1, 2 and 3 have been compared, additional experiment of PARP depletion have been included, and a proof-of-concept siRNA screen has been included. To my opinion, the manuscript in its current form better underscores the potential use of the technique. Also the phrasing in the abstract and body text better fits the results. In its present form, I think the manuscript is acceptable for publication.

We are grateful to this reviewer for insightful comments on our manuscript and for supporting publication of the revised paper.

Reviewer #2 (Remarks to the Author):

The authors have done a fine job and paper should be accepted

We are also thankful to this reviewer for constructive comments and for supporting publication of our work.

Reviewer #3 (Remarks to the Author):

The authors have adequately addressed my questions. The manuscript has been extensively revised and much improved. It would be better if the authors are able to discuss the DNA lesions induced by PARP inhibitors in S phase, which might be helpful for others using these inhibitors for basic research as well as clinical treatment for cancer patients.

We were glad to read that this reviewer feels that the revised manuscript has improved significantly. As suggested, we added a paragraph in the discussion on the types of DNA lesions induced by PARP inhibitors in S phase (pages 21/22).

Reviewer #5 (Remarks to the Author):

Michelena et al. have addressed all of the original points raised in the previous review in great detail and provided a substantial amount of new data that significantly improves the manuscript. In particular, the authors identify a novel synthetic lethal interaction between PARPi and pevonedistat in a new targeted high-content imaging screen using

QIBC. However, there are still some outstanding issues that should be addressed before publication:

We are grateful to this reviewer for having evaluated our revised manuscript and for acknowledging the additional data and the improvements. We addressed all outstanding issues as detailed below.

- As this is a technology manuscript the authors should provide much more detailed information in the Material and Methods section to allow other laboratories to follow their protocols. This is especially relevant when it comes to the QIBC analysis. E.g. while the color coding of the scatter dot blots results in easily understandable and beautifully presented Figure panels, it is not very accurate. For example in Figure 1b red dots are supposedly corresponding to cells with a γ H2AX “value” of 200. However, when looking at the blot, red dots can be found well below the 200 intensity on the y-axes (it is probably rather a range value). How exactly was the data plotted and the final graphs generated? How were the RAD51 foci counted? Furthermore, there is no information on the new targeted high-content imaging screen in the Material and Methods section and on how the 96 well and 384 well plate immunostaining experiments were performed.

We are in the process of preparing a step-by-step protocol for upload to Nature Protocol Exchange. In addition, we have also extended the Methods section of our manuscript to include more detailed information on the siRNA transfection in multi-well plates and for the immunostaining procedure in this screening format. The data plots and the final graphs are generated using ScanR and Spotfire software as stated in the Methods section. In our step-by-step protocol we mention alternative software packages to achieve comparable results. In addition, we now provide an extended Supplementary Figure 2a, which shows how RAD51 foci are detected in our assays. This figure should help other researchers to compare their RAD51 staining and RAD51 foci detection masks. The color code in Figure 1b, as spotted by the reviewer, contained a mistake, and we are very grateful to the reviewer for the careful inspection of the data. A derived parameter different from the mean γ H2AX intensity per nucleus was accidentally displayed and caused the discrepancy when comparing the color code to the y-axis values. We corrected this mistake and in the new Figure 1b the color code now correlates with the y-axis values. We apologize for this oversight.

- In the newly added target siRNA-based screen, the authors show that TIMELESS knock down leads to a dramatic increase in PARPi sensitivity. Besides being involved in the replication checkpoint, TIMELESS has been recently shown to interact with PARP1. The authors should briefly discuss this in their manuscript.

We thank the reviewer for pointing this out and included references to the interaction between PARP1 and TIMELESS (Xie et al. Mol Cell 2015; Young et al. Cell Rep 2015).